# Error Feedback for Smooth and Nonsmooth Convex Optimization with Constant, Decreasing and Polyak Stepsizes

## Abstract

Error feedback, originally proposed a decade ago by Seide et al (2014), is an immensely popular strategy for stabilizing the convergence behavior of distributed algorithms employing communication compression via the application of contractive compression operators, such as greedy and random sparsification, quantization, and low-rank approximation. While our algorithmic and theoretical understanding of error feedback has grown immensely over the years, several important considerations remained elusive. For example, the theory of error feedback is fully focused on the smooth convex and nonconvex regimes, and results in the nonsmooth convex setting are limited. This is not a coincidence: Error feedback works when the gradients converge, and this is not necessarily the case in the nonsmooth setting. Further, existing stepsize rules for error feedback are limited to constant schedules; a by-product of the current theoretical approach to analyzing error feedback. By modifying the algorithmic design of error feedback, we are able to resolve these issues. In particular, we provide a comprehensive analysis covering both the smooth and nonsmooth convex regimes, and give support for constant, decreasing and adaptive (Polyak-type) stepsizes. This is the first time such results are obtained. In particular, this is the first time adaptive stepsizes have successfully been combined with compression mechanisms. Our theoretical results are corroborated with suitable numerical experiments.

## 1 Introduction

Machine learning tasks can be formulated into the finite-sum optimization:

$$\min_{x \in \mathbb{R}^d} f(x), \tag{1}$$

where $f(x) = (1/n)\sum_{i=1}^{n} f_i(x)$ is a convex, but not necessarily smooth function, $n$ is the number of training data samples, and $f_i(x)$ is the loss of the model vector $x \in \mathbb{R}^d$ on data sample $i$. In this paper, we focus on the problem of minimizing the function that preserves the property of convexity and the existence of minimizer.

**Assumption 1.** *The function $f$ is convex.*

**Assumption 2.** *The function $f$ has at least one minimizer, denoted by $x_\star$.*

Classical instances of problem (1) include empirical risk minimization for supervised learning tasks. In these tasks, each component function $f_i(x) = \phi(\langle a_i, x \rangle, b_i)$, where $a_i \in \mathbb{R}^d$ is the $i^{\text{th}}$ data sample vector with its associated output $b_i \in \mathbb{R}$. Here, $\phi : \mathbb{R} \times \mathbb{R} \to \mathbb{R}$ is a loss function that measures how close the predicted output $\langle a_i, x \rangle$ is to the true output $b_i$. Examples of loss functions include mean squared error loss in the least squares problems (Allen, 1971), cross-entropy loss in neural network training (Zhang & Sabuncu, 2018), and hinge loss in support vector machines (Hearst et al., 1998).

Substantial attention has been directed towards developing optimization algorithms to solve machine learning problems within desirable solution accuracy and training time. Among simple optimization algorithms is stochastic gradient descent (SGD), which can be implemented under distributed computing environments that leverage multiple devices to train the learning model. However, the

increasing learning model size to demonstrate impressive classification, recognition, and reasoning capabilities poses significant challenges. This leads to immense resources in computation, storage, and energy for running the algorithm. For example, the ResNet-50 (He et al., 2016) model comprises 23 million parameters, which needs 95 MB for storage and 4 GFLOPs (Giga Floating Point Operations) of the computations (You et al., 2019). Training ResNet-50 takes 14 days on one NVIDIA M40 GPU (You et al., 2019). These advanced models, therefore, place immense strain on the communication between workers and servers during algorithm execution, underscoring the need for optimization algorithms that can minimize communication overhead especially when training across communication-constrained networks of edge devices.

To alleviate communication requirements, one common approach is to compress model parameters or gradients before they are used in optimization algorithms. Widely used compressors encompass *sparsification* (Alistarh et al., 2018; LeCun et al., 1989; Hagiwara, 1993) where a few important vector elements are kept, *quantization* (Alistarh et al., 2017; Wen et al., 2017; Bernstein et al., 2018) where each vector element is mapped into a smaller set of discrete finite values, and *low-rank approximation* (Vogels et al., 2019; Safaryan et al., 2022a) that reduces the number of neural network parameters. Empirical observations suggest significant communication savings for running optimization algorithms that utilize compressors. However, these algorithms suffer from poor convergence performance or even diverge. Distributed gradient algorithms with biased compressors may diverge exponentially for strongly convex quadratic functions (Beznosikov et al., 2023).

## 1.1 ERROR FEEDBACK

Error feedback (EF), originally proposed by Seide et al. (2014) a decade ago, is a mechanism that stabilizes compressed optimization algorithms. Several EF variants were studied in the centralized or distributed setting by, e.g., Stich et al. (2018); Cordonnier (2018); Wu et al. (2018); Alistarh et al. (2018); Karimireddy et al. (2019); Gorbunov et al. (2020); Qian et al. (2021); Chen et al. (2021); Basu et al. (2019); Khirirat et al. (2020). However, analyses for distributed EF algorithms often make the bounded gradient (Cordonnier, 2018; Alistarh et al., 2018) or homogeneous (IID) data assumption (Stich & Karimireddy, 2019), thus leading to the less optimal rate (e.g. $\mathcal{O}(1/K^{2/3})$ for nonconvex problems). To alleviate these issues, one re-engineered EF mechanism, called EF21, was proposed by Richtárik et al. (2021). EF21 attains the $\mathcal{O}(1/K)$ rate in the deterministic regime (Richtárik et al., 2021) with the provably lower complexity constant than classical gradient descent for sparse problems (Richtárik et al., 2023), and the $\mathcal{O}(1/K^{1/2})$ rate in the stochastic setting (Fatkhullin et al., 2024). Another recent EF variant that enjoys fast convergence over arbitrarily heterogeneous data is EControl (Gao et al., 2023).

## 1.2 NONSMOOTH OPTIMIZATION

The convergence bounds of error feedback algorithms have been extensively studied under the smoothness assumption of objective functions, such as Lipschitz continuity of the gradients. However, this assumption often fails to capture machine learning problems that involve minimizing nonsmooth functions, e.g. Lipschitz continuity of the functions. For example, support vector machines (Hearst et al., 1998) utilize hinge loss to achieve sparse solutions, while robust regression (Holland & Welsch, 1977) employs absolute loss to create regressors that are resistant to outliers. Moreover, many deep neural networks exhibit nonsmooth behaviors, particularly when using activation functions such as rectified linear units (ReLU) and leaky ReLU (Maas et al., 2013). In various network models, the Lipschitz continuity parameter of the objective functions can be estimated efficiently, as demonstrated in, e.g. Anil et al. (2019); Béthune et al. (2023); Jordan & Dimakis (2020); Latorre et al. (2020).

Designing optimization methods for nonsmooth objective functions possess significant challenges, compared to optimizing smooth functions. Subgradient methods are commonly used to tackle nonsmooth problems. However, when these methods incorporate compressed subgradients to reduce communication costs, they may not ensure necessarily convergence. This issue is particularly evident in the case of signSGD which can diverge even for simple nonsmooth problems by Karimireddy et al. (2019). To mitigate this problem, a very limited number of prior works have explored the benefits of error feedback in nonsmooth regimes. In particular, Karimireddy et al. (2019) demonstrated that error feedback can enable subgradient methods to achieve the $\mathcal{O}(1/\sqrt{K})$ convergence rate while

reducing communication costs. Nonetheless, the existing analysis frameworks requires the stepsizes to be constant, which may not yield optimal convergence performance in practice. This raises an open question: *can we develop a convergence analysis framework that allows error feedback algorithms to accomodate parameter-free stepsize rules under possibly nonsmooth regimes?*

## 1.3 CONTRIBUTIONS

We summarize our main contributions as follows:

**EF21-P for nonsmooth and smooth convex regimes.** In contrast to Gruntkowska et al. (2023), which analyzes EF21-P only using deterministic gradients and constant step sizes, we investigate EF21-P using subgradients (either deterministic or stochastic) and various step size strategies in both nonsmooth and smooth convex regimes. EF21-P serves as a useful error feedback algorithm in the primal space of models. It can be combined with other appropriate algorithms to design communication-efficient distributed algorithms, and it encompasses many first-order algorithms of interest, such as EF14 and primal averaging algorithms.

**Comprehensive convergence analysis for any stepsize rules under standard assumptions.** We revisit the convergence analysis of EF21-P in both nonsmooth and smooth convex regimes. Our analysis introduces a novel descent inequality that explicitly captures the impact of any stepsize rules on the convergence behaviors of EF21-P using either deterministic or stochastic subgradients. This inequality is derived under standard assumptions on compressors and objective functions. While compressors are assumed to be $\alpha$-contractive, objective functions are assumed to be convex, posses at least one minimizer, and have Hölder continuous subgradients.

**Sublinear rate for constant, decreasing, and Polyak stepsizes.** We demonstrate how to apply our descent inequality to derive the convergence rates of EF21-P using three common step size strategies, i.e. constant, decreasing, and Polyak, in both smooth and nonsmooth convex regimes. Our approach contrasts with most existing works, which typically analyze error feedback algorithms only in smooth regimes. To the best of our knowledge, Karimireddy et al. (2019) is the only work that provides convergence results for EF14 in both regimes; however, their analysis is restricted to constant step sizes. A summary of the theoretical comparisons between existing results and our findings for error feedback algorithms is provided in Table 1. Additionally, we derive the sublinear rates of EF21-P. With decreasing stepsizes of the form $\gamma_k = \gamma_0/\sqrt{k+1}$ for $k \in [0, K]$ and some positive scalars $\gamma_0$, EF21-P achieves a $\mathcal{O}(\log(K+1)/\sqrt{K})$ rate in nonsmooth regimes and a $\mathcal{O}(1/\sqrt{K})$ rate in smooth regimes. With constant and Polyak stepsizes, EF21-P achieves a $\mathcal{O}(1/\sqrt{K})$ rate in nonsmooth regimes and $\mathcal{O}(1/K)$ rate in smooth regimes.

**The first successful combination of adaptive stepsizes with error feedback algorithms.** Unlike existing works that are limited to constant or decreasing stepsizes, our finding marks the first successful application of the adaptive Polyak stepsizes to first-order algorithms that use error-feedback compression. Specfically, for both nonsmooth and smooth regimes, EF21-P with the Polyak stepsize attains the same convergence rate as that with the carefully tuned constant stepsize. The Polyak stepsize, unlike the constant stepsize, does not need to know smoothness parameters, which are often inaccessible in practice.

**Numerical evaluations.** We verify our theory via numerical experiments on EF21-P for solving logistic regression problems over synthetic data and support vector machines problems over LIB-SVM data (Chang & Lin, 2011), which represent smooth and nonsmooth convex problems. Our results confirm that by admitting larger learning rates, the Polyak stepsize outperforms constant and decreasing stepsizes.

## 1.4 NOTATIONS

We use the following notations throughout this paper. For $x, y \in \mathbb{R}^d$, $\langle x, y \rangle := x^T y$ is the inner product, and $\|x\| := \sqrt{\langle x, x \rangle}$ is the Euclidean norm. For a real-valued function $f : \mathbb{R}^d \to \mathbb{R}$, $f(x_\star)$ is its minimum with a minimizer $x_\star = \arg\min_{x \in \mathbb{R}^d} f(x)$.

| Method | Stepsize | Smooth | Nonsmooth | Note |
|:------:|:--------:|:------:|:---------:|:----:|
| EF21
Richtárik et al. (2021) | Fixed | ✓ | ✗ | nCVX and $\mu$-PL nonconvexity |
| EF21-P
Gruntkowska et al. (2023) | Fixed | ✓ | ✗ | sCVX and nCVX |
| EControl
Gao et al. (2023) | Fixed | ✓ | ✗ | sCVX, CVX, nCVX |
| EF14
Gorbunov et al. (2020) | Fixed | ✓ | ✗ | sCVX, CVX |
| EF14
Stich et al. (2018) | Decreasing | ✓ | ✗ | sCVX + Bounded gradient norm |
| EF14
Stich & Karimireddy (2020) | Fixed | ✓ | ✗ | sCVX, CVX, nCVX |
| EF14
Beznosikov et al. (2023) | Fixed,
Decreasing | ✓ | ✗ | sCVX |
| EF14
Karimireddy et al. (2019) | Fixed | ✓ | ✓ | nCVX for smooth and CVX for nonsmooth |
| EF21-P
This paper | Polyak,
Fixed,
Decreasing | ✓ | ✓ | CVX |

Table 1: Known and our results for error feedback algorithms. EF21-P is equivalent to EF14, as shown by Gruntkowska et al. (2023). Here, sCVX, CVX and nCVX are strongly convex, convex and nonconvex problems, respectively. In this paper, we provide the comprehensive convergence analysis of EF21-P with constant, decreasing, and Polyak stepsizes for both nonsmooth and smooth convex problems. Notably, we provide the first result showing how adaptive Polyak stepsizes can be integrated into error feedback algorithms. Our approach significantly differs from previous works that have only examined error feedback algorithms with constant and/or decreasing stepsizes mostly for smooth problems.

## 2 RELATED WORK

**Stepsize selection.** The main hyperparameter that affects the convergence of stochastic optimization algorithms such as SGD is the stepsize. Several stepsizes have been proposed to guarantee and improve the algorithmic convergence. For example, SGD converges towards the neighborhood of the minimizer when we use a constant stepsize (Gower et al., 2019; Garrigos & Gower, 2023; Gower et al., 2021; Ghadimi & Lan, 2013; Needell et al., 2014), and it enjoys the convergence towards the exact minimizer when we choose a decreasing stepsize (Gower et al., 2019; Garrigos & Gower, 2023; Shamir & Zhang, 2013; Robbins & Monro, 1951) and step-decay stepsize (Ge et al., 2019; Wang et al., 2021; Li & Arora, 2020). However, these stepsize schedules require significant hyperparameter hand-tuning efforts to maximize the convergence speed in practice.

**Polyak stepsizes.** Adaptive stepsize rules adjust stepsizes to maximize the algorithmic convergence on the fly. One classical adaptive stepsize rule is by Polyak (1987). The Polyak stepsize computes its value based on the function values at the current iterate and the minimizer, and on the Euclidean norm of the (sub)gradient to maximize a convergence bound at each iteration. Well-known results of deterministic (sub)gradient descent with the Polyak stepsize were shown by Hazan & Kakade (2019); Boyd et al. (2003). The algorithm attains $\mathcal{O}(1/K)$ convergence for smooth convex regimes and the $\mathcal{O}(1/K^{1/2})$ convergence for nonsmooth convex regimes. The Polyak stepsize has also been extended to stochastic optimization algorithms, such as SGD and stochastic momen-

tum algorithms, by many recent works including Prazeres & Oberman (2021); Berrada et al. (2020); Loizou et al. (2021); Jiang & Stich (2024); Schaipp et al. (2023); Wang et al. (2023).

**Other adaptive stepsizes.** Other adaptive stepsize schedules are line search and statistical adaptive procedures. First, line search techniques determine an acceptable stepsize at each iteration to obtain a sufficient decrease in the objective function. One common line search is Armijo Armijo (1966). Convergence analyses of gradient-based algorithms using the Armijo line search and its variants have been conducted in both deterministic and stochastic settings, as demonstrated by Vaswani et al. (2019); Ahookhosh & Ghaderi (2017); Galli et al. (2024); Kunstner et al. (2024); Jiang & Stich (2024). Second, statistical tests detect stationarity of the iterate progress to decrease the fixed stepsize. Despite their convergence benefits empirically by Zhang et al. (2020); Pesme et al. (2020), the algorithms using these adaptive tests lack theoretical guarantees.

**Adaptive gradient methods.** Adaptive gradient algorithms refer to SGD that use stepsizes computed adaptively based on magnitudes of previous stochastic gradients. Popular adaptive gradient algorithms include Adagrad (Duchi et al., 2011), Adam (Kingma & Ba, 2015) and RMSProp (Tieleman, 2012) for effectively solving sparse optimization (Duchi et al., 2013) and deep learning tasks (LeCun et al., 2015).

## 3 EF21-P: USEFUL FORM FOR ERROR FEEDBACK ALGORITHMS

EF21-P (Gruntkowska et al., 2023) is an algorithm that solves problem (1) by performing error-feedback updates in the primal space of the models/iterates. Given the initial iterates $x_0, w_0 \in \mathbb{R}^d$, and the compressor $\mathcal{C}_k$, EF21-P updates the iterates $x_k, w_k$ via:

$$x_{k+1} = x_k - \gamma_k g(w_k), \quad \text{and} \quad w_{k+1} = w_k + \mathcal{C}_k(x_{k+1} - w_k). \tag{2}$$

Here $\gamma_k$ is any positive stepsize, and $g(w_k)$ is either the subgradient of $f$ at $w_k$ in the deterministic setting or the subgradient of $f_{i_k}$ at $w_k$, where $i_k$ is sampled from $\{1, 2, \ldots, n\}$ uniformly at random in the stochastic setting. See the full description of EF21-P in Algorithm 1.

---
**Algorithm 1** Primal variant of Error Feedback 2021 (EF21-P)

---
1: **Parameters:** Starting points $x_0, w_0 \in \mathbb{R}^d$; learning rates $\gamma_k > 0$ for $k = 0, 1, \ldots$; $\alpha$-contractive compressors $\mathcal{C}_k : \mathbb{R}^d \to \mathbb{R}^d$ for $k = 0, 1, \ldots$
2: **for** $k = 0, 1, 2, \ldots$ **do**
3:     Set $g(w_k)$ to be a subgradient of $f$ at $w_k$ (**deterministic setting**), or a subgradient of $f_{i_k}$ at $w_k$ where $i_k$ is selected uniformly at random from $\{1, 2, \ldots, n\}$ (**stochastic setting**).
4:     $x_{k+1} = x_k - \gamma_k g(w_k)$
5:     $w_{k+1} = w_k + \mathcal{C}_k(x_{k+1} - w_k)$
6: **end for**

---

Furthermore, EF21-P is a very useful update for error feedback algorithms for four main reasons. First, EF21-P is equivalent to the traditional EF14 algorithms by Seide et al. (2014). By taking $e_k := x_k - w_k$, the EF21-P update in (2) can be expressed equivalently as

$$w_{k+1} = w_k + \mathcal{C}_k(e_k - \gamma_k g(w_k)), \quad \text{and} \quad e_{k+1} = e_k - \gamma_k g(w_k) - \mathcal{C}_k(e_k - \gamma_k g(w_k)). \tag{3}$$

Second, EF21-P can be combined with suitable algorithms, e.g. DIANA and DCGD, to obtain distributed algorithms that utilize bi-directional compression to improve communication complexity (Gruntkowska et al., 2023). Third, EF21-P yields stochastic primal averaging algorithms (Defazio, 2020; Tao et al., 2018) when $\mathcal{C}_k(v) = \alpha_k \cdot v$ for $\alpha_k \in (0, 1]$ in (2). Fourth, EF21-P recovers (stochastic) subgradient descent when the compressors $\{\mathcal{C}_k\}$ are chosen to be identity operators.

To facilitate our convergence analysis of EF21-P that accommodates any stepsize rules, we impose Assumptions 3 and 4 on the objective function and the compressor, respectively.

**Assumption 3.** *The function $f$ has Hölder continuous subgradient. That is, there exists $L > 0$ and $\eta \in [0, 1]$ such that its subgradient $g(x)$ satisfies*

$$\|g(x) - g(y)\| \le L_\eta \|x - y\|^\eta, \quad \forall x, y \in \mathbb{R}^d.$$

On the one hand, Assumption 3 with $\alpha = 1$ implies that the function $f$ has an $L_1$-Lipschitz continuous gradient. On the other hand, Assumption 3 with $\eta = 0$ is equivalent to the subgradients of the function $f$ that have the norms uniformly bounded by $G \geq L_0/2$. This $G$-bounded norm of the subgradient is equivalent to the $G$-Lipschitz continuity of the functions, according to Lemma 8.8. of Garrigos & Gower (2023).

**Assumption 4** ($\alpha$-contractiveness). *The compressors $\{\mathcal{C}_k\}_{k \geq 0}$ are $\alpha$-contractive. That is, there exists $\alpha \in (0, 1]$ such that*

$$\mathrm{E}\left[\|\mathcal{C}(x) - x\|^2\right] \leq (1 - \alpha)\|x\|^2, \qquad \forall x \in \mathbb{R}^d. \tag{4}$$

## 4  ANALYSIS FOR CONVEX, DETERMINISTIC OPTIMIZATION

We begin by studying EF21-P for convex, deterministic optimization. The result is of importance for designing distributed algorithms that use bi-directional compression based on the combination of EF21-P and other suitable algorithms (see the discussion Section 3). To derive the results for popular stepsize schedules, we develop a novel key descent inequality below.

**Lemma 1** (**Convex, deterministic optimization**). *Consider problem (1), where $f(x)$ satisfies Assumption 1 (convexity), and Assumption 2 (existence of a minimizer). Also, let Assumption 4 (contractivity) hold, and let the stepsizes $\{\gamma_k\}$ be constants conditional on $x_k$ and $w_k$. Then the iterates $\{x_k, w_k\}$ generated by Algorithm 1 (EF21-P), where $g(w_k)$ is the subgradient of $f$ at $w_k$ satisfy the inequality/recursion*

$$\mathrm{E}\left[V_{k+1}|\, x_k, w_k\right] \leq V_k - 2\gamma_k\left(f(w_k) - f(x_\star)\right) + B\gamma_k^2\|g(w_k)\|^2,$$

*where the Lyapunov function $V_k$ and constants $B$ are given by*

$$V_k := \|x_k - x_\star\|^2 + \frac{1}{\lambda\left(1 - (1-\alpha)(1+\theta)\right)}\|w_k - x_k\|^2, \text{ and } B = 1 + \lambda + \frac{1}{\lambda}\frac{(1-\alpha)(1+1/\theta)}{1 - (1-\alpha)(1+\theta)}.$$

*Above, $\lambda > 0$ can be chosen arbitrarily, and $\theta > 0$ is any constant such that $(1-\alpha)(1+\theta) \in (0, 1)$.*

Lemma 1 implies a decrease in the Lyapunov function $V_k$, when the stepsizes $\gamma_k$ are chosen to ensure that $-2\gamma_k\left(f(w_k) - f(x_\star)\right) + B\gamma_k^2\|g(w_k)\|^2 < 0$ for all $k$. The rate of decrease in $V_k$ depends on the stepsizes $\gamma_k$, the compression parameter $\alpha \in (0, 1]$, and two free parameters due to Young's inequality $\lambda, \theta > 0$. Also, from this lemma, the descent inequality of EF21-P with $\mathcal{C}(x) \equiv x$ (i.e., $\alpha = 1$) implies

$$\|x_{k+1} - x_\star\|^2 \leq \|x_k - x_\star\|^2 - 2\gamma_k\left(f(x_k) - f(x_\star)\right) + (1 + \lambda)\gamma_k^2\|g(x_k)\|^2.$$

Due to the presence of $\lambda > 0$, this inequality is worse than that of the classical subgradient method:

$$\|x_{k+1} - x_\star\|^2 \leq \|x_k - x_\star\|^2 - 2\gamma_k\left(f(x_k) - f(x_\star)\right) + (1 + 0)\gamma_k^2\|g(x_k)\|^2.$$

**Optimal choices of $\lambda, \theta > 0$.**   Although we are free to choose $\lambda, \theta > 0$, we wish to select $\lambda, \theta$ that minimizes $B$ in the last term on the right-hand side of the inequality from Lemma 1. We achieve this by first choosing $\theta_\star = \frac{1}{\sqrt{1-\alpha}} - 1$ to minimize $\frac{(1-\alpha)(1+1/\theta)}{1-(1-\alpha)(1+\theta)}$ such that $(1-\alpha)(1+\theta_\star) \in (0, 1)$ from Lemma 3 of Richtárik et al. (2021), and second by setting $\lambda_\star = \frac{\sqrt{1-\alpha}}{1-\sqrt{1-\alpha}}$ to minimize $B$. The associated minimal value of $B$ is $B_\star = 1 + 2\lambda_\star$.

## 5  THEOREMS FOR CONSTANT, DECREASING AND POLYAK STEPSIZES

Now, we demonstrate how the descent lemma from Section 4 can be applied to establish the convergence rate results for EF21-P that uses three main stepsize schedules: constant, decreasing, and Polyak stepsizes. Our results apply for both nonsmooth and smooth convex regimes, and the optimal rates are summarized in Table 2. Derivations of all results can be found in the appendix.

| stepsize $\gamma_k$ | Assumption 3 with $\alpha = 0$ | Assumption 3 with $\alpha = 1$ | rate | required knowledge of hyper-parameters |
|---|---|---|---|---|
| Constant (Theorem 1) | ✓ | ✗ | $\frac{G\sqrt{B}}{\sqrt{K}}\sqrt{V_0}$ | $G, B, K, V_0$ |
| Constant (Theorem 1) | ✗ | ✓ | $\frac{2L_1 B}{K}V_0$ | $L_1, B$ |
| Decreasing (Theorem 2) | ✓ | ✗ | $\mathcal{O}\left(\frac{\log(K+1)}{\sqrt{K}}\right)$ | ✗ |
| Decreasing (Theorem 2) | ✗ | ✓ | $\mathcal{O}\left(\frac{1}{\sqrt{K}}\right)$ | $L_1, B$ |
| Polyak (Theorem 3) | ✓ | ✗ | $\frac{G\sqrt{B}}{\sqrt{K}}\sqrt{V_0}$ | $B$ |
| Polyak (Theorem 3) | ✗ | ✓ | $\frac{2L_1 B}{K}V_0$ | $B$ |

Table 2: Convergence rates of EF21-P with constant, decreasing and Polyak stepsizes for convex, deterministic optimization. Here, $V_0, B$ are defined in Lemma 1. For nonsmooth problems, $f$ has the norm of its subgradient upper-bounded by $G \geq L_0/2$. For smooth problems, $f$ has an $L_1$-Lipschitz continuous gradient. The $\mathcal{O}(1/\sqrt{K})$ and $\mathcal{O}(1/K)$ convergence rate can be attained by constant and Polyak stepsizes, unlike decreasing stepsizes. Polyak stepsizes, in contrast to constant stepsizes, do not require to know Lipschitz parameters that are often inaccessible.

## 5.1 CONSTANT STEPSIZE

First, we provide the convergence for EF21-P with constant stepsizes, which are commonly used to analyze optimization algorithms.

**Theorem 1** (**Constant stepsize for deterministic optimization**). *Consider the iterates $\{w_k\}$ generated by Algorithm 1 (EF21-P), where $g(w_k)$ is the subgradient of $f$ at $w_k$, for solving problem (1). Let the assumptions invoked in Lemma 1 hold, and choose the stepsize*

$$\gamma_k = \gamma > 0.$$

*Here, $B = 1 + \lambda + \frac{1}{\lambda}\frac{(1-\alpha)(1+1/\theta)}{1-(1-\alpha)(1+\theta)}$, $\lambda > 0$ is arbitrarily chosen, and $\theta > 0$ is any constant such that $(1-\alpha)(1+\theta) \in (0,1)$.*

1. (**Nonsmooth case**). *If $f$ satisfies Assumption 3 with $\alpha = 0$ ($G$-bounded norm of $g(w)$ with $G \geq L_0/2$), then for any $\gamma > 0$,*

$$\mathrm{E}\left[f\left(\frac{1}{K}\sum_{k=0}^{K-1}w_k\right) - f(x_\star)\right] \leq \frac{V_0}{2\gamma K} + \frac{BG^2\gamma}{2}.$$

2. (**Smooth case**). *If $f$ satisfies Assumption 3 with $\alpha = 1$ ($L_1$-Lipschitzness of $\nabla f$) holds, then for any $0 < \gamma < 1/(BL_1)$,*

$$\mathrm{E}\left[f\left(\frac{1}{K}\sum_{k=0}^{K-1}w_k\right) - f(x_\star)\right] \leq \frac{V_0}{2\gamma\left(1 - BL_1\gamma\right)K}.$$

From Theorem 1, EF21-P with the constant stepsize $\gamma_k = \gamma$ achieves the $\mathcal{O}(1/K)$ convergence. For nonsmooth problems, EF21-P attains the convergence up to the additive constant $BG^2\gamma/2$ for any $\gamma > 0$. For smooth problems, EF21-P converges for $0 < \gamma < 1/(BL_1)$. Our results hold for any initial iterates $w_0, w_0 \in \mathbb{R}^d$, unlike Theorem D.1. and E.3. of Gruntkowska et al. (2023) that assumes $w_0 = x_0$. Additionally, we can select $\gamma > 0$ to minimize the convergence bounds for both problems. On the one hand, for nonsmooth problems, if we choose $\gamma_\star = \arg\min_{\gamma > 0}\left(\frac{V_0}{2\gamma K} + \frac{BG^2\gamma}{2}\right) = \frac{1}{\sqrt{K}}\sqrt{\frac{V_0}{BG^2}}$, then

$$\mathrm{E}\left[f(\hat{w}_K) - f(x_\star)\right] \leq \frac{G\sqrt{B}}{\sqrt{K}}\sqrt{V_0}.$$

On the other hand, for smooth problems, if we select $\gamma_\star = \arg\min_{\gamma > 0}\gamma(1 - BL_1\gamma) = \frac{1}{2BL_1}$, then

$$\mathrm{E}\left[f(\hat{w}_K) - f(x_\star)\right] \leq \frac{2BL_1V_0}{K}.$$

A primary drawback of employing constant stepsizes is the necessity for knowledge concerning the number of iteration counts $K$, and two often inaccessible parameters: the Lipschitz parameters $L_0, L_1$, and the Lyapunov function at the initial iterates $V_0$ (given $x_\star, w_0, x_0$). Later, we show that such rates achieved by these constant stepsizes can also be attained by the Polyak stepsize without requiring these parameters in Section 5.3.

## 5.2 DECREASING STEPSIZE

Second, we present the convergence of EF21-P with decreasing step-sizes.

**Theorem 2** (**Decreasing stepsize for deterministic problems**). *Consider the iterates $\{w_k\}$ generated by Algorithm 1* (EF21-P), *where $g(w_k)$ is the subgradient of $f$ at $w_k$, for solving problem (1). Let the assumptions invoked in Lemma 1 hold, and choose the stepsize*

$$\gamma_k = \frac{\gamma_0}{\sqrt{k+1}} \quad \text{with} \quad \gamma_0 > 0.$$

*Here, $B = 1 + \lambda + \frac{1}{\lambda}\frac{(1-\alpha)(1+1/\theta)}{1-(1-\alpha)(1+\theta)}$, $\lambda > 0$ is arbitrarily chosen, and $\theta > 0$ is any constant such that $(1-\alpha)(1+\theta) \in (0,1)$.*

1. (**Nonsmooth case**). *If $f$ satisfies Assumption 3 with $\alpha = 0$ (G-bounded norm of $g(w)$ with $G \geq L_0/2$), then*

$$\mathrm{E}\left[f\left(\frac{1}{\sum_{k=0}^{K-1}\gamma_k}\sum_{k=0}^{K-1}\gamma_k w_k\right) - f(x_\star)\right] \leq \frac{V_0 + (2BG^2)\gamma_0^2\log(K+1)}{\gamma_0\sqrt{K}}.$$

2. (**Smooth case**). *If $f$ satisfies Assumption 3 with $\alpha = 1$ ($L_1$-Lipschitzness of $\nabla f$) holds, and $\gamma_0 \in (0, 1/(2BL_1)]$, then*

$$\mathrm{E}\left[f\left(\frac{1}{\sum_{k=0}^{K-1}\gamma_k}\sum_{k=0}^{K-1}\gamma_k w_k\right) - f(x_\star)\right] \leq \frac{2V_0}{\gamma_0\sqrt{K}}.$$

From Theorem 2, the decreasing stepsizes guarantee the EF21-P convergence at the $\mathcal{O}(\log(K+1)/\sqrt{K})$ rate for nonsmooth problems, and at the $\mathcal{O}(1/\sqrt{K})$ rate for smooth problems. These rates however are slower than constant stepsizes from Section 5.1.

## 5.3 POLYAK STEPSIZE

Third, inspired by Polyak (1987), we show how to obtain the Polyak stepsize by using Lemma 1, and applying the derivation provided by Hazan & Kakade (2019) for EF21-P. The subsequent result establishes the convergence attained by the Polyak stepsize.

**Theorem 3** (**Polyak stepsize for deterministic problems**). *Consider the iterates $\{w_k\}$ generated by Algorithm 1* (EF21-P), *where $g(w_k)$ is the subgradient of $f$ at $w_k$, for solving problem (1). Let the assumptions invoked in Lemma 1 hold, and choose the stepsize*

$$\gamma_k = \frac{f(w_k) - f(x_\star)}{B\|g(w_k)\|^2}.$$

*Here, $B = 1 + \lambda + \frac{1}{\lambda}\frac{(1-\alpha)(1+1/\theta)}{1-(1-\alpha)(1+\theta)}$, $\lambda > 0$ is arbitrarily chosen, and $\theta > 0$ is any constant such that $(1-\alpha)(1+\theta) \in (0,1)$.*

1. (**Nonsmooth case**). *If $f$ satisfies Assumption 3 with $\alpha = 0$ (G-bounded norm of $g(w)$ with $G \geq L_0/2$), then*

$$\mathrm{E}\left[f\left(\frac{1}{K}\sum_{k=0}^{K-1}w_k\right) - f(x_\star)\right] \leq \frac{G\sqrt{B}}{\sqrt{K}}\sqrt{V_0}.$$

2. (**Smooth case**). *If $f$ satisfies Assumption 3 with $\alpha = 1$ ($L_1$-Lipschitzness of $\nabla f$) holds, then*

$$\mathrm{E}\left[f\left(\frac{1}{K}\sum_{k=0}^{K-1}w_k\right) - f(x_\star)\right] \leq \frac{2L_1B}{K}V_0.$$

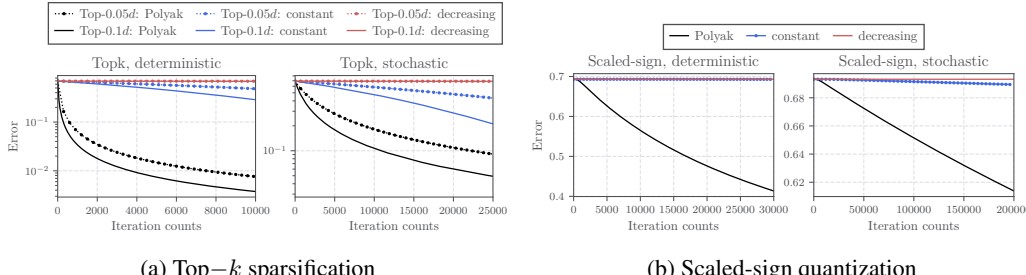

(a) Top$-k$ sparsification          (b) Scaled-sign quantization

Figure 1: Convergence $f(w_k) - f(x_\star)$ w.r.t. iteration $k$ of EF21-P in the deterministic and stochastic setting for logistic regression over synthetic data with $(n, d) = (1000, 10000)$ generated by Nutini et al. (2022).

**First known result for combining Polyak stepsize with error feedback.** This theorem is the first known result for EF21-P that uses the Polyak stepsize. This stepsize achieves the same convergence rate for EF21-P as the constant stepsize $\gamma_\star = \frac{1}{\sqrt{K}} \sqrt{\frac{V_0}{BG^2}}$ for nonsmooth problems, and as $\gamma_\star = \frac{1}{2BL_1}$ for smooth problems. Unlike these constant stepsizes, the Polyak stepsize does not require Lipschitz parameters, except $f(x_\star)$. Nonetheless, $f(x_\star)$ can be estimated by its lower-bound (Hazan & Kakade, 2019), which is often zero for unregularized logistic regression problems, and other problems that satisfy the interpolation condition, as discussed later in Section 6.

## 6 ANALYSIS FOR CONVEX, STOCHASTIC OPTIMIZATION UNDER INTERPOLATION

Next, we turn our attention to EF21-P for convex, stochastic optimization under interpolation. We rather consider EF21-P that evaluates the subgradient of $f_{i_k}$ at $w_k$, where $i_k$ is selected uniformly at random from $\{1, 2, \ldots, n\}$. To facilitate the analysis, we further assume the interpolation condition, which holds when problem (1) is the problem of training an over-parameterized model, such as a deep neural network, solving a consistent linear system, and learning a classifier over linearly separable data, (Loizou et al., 2021).

**Assumption 5.** *(Interpolation) Let $x_\star = \arg\min_{x \in \mathbb{R}^d} f(x) := \frac{1}{n} \sum_{i=1}^{n} f_i(x)$. Then, $\nabla f_i(x_\star) = 0$ for $i = 1, 2, \ldots, n$.*

To this end, we prove the descent inequality for EF21-P in the stochastic setting.

**Lemma 2** (**Convex, stochastic optimization under interpolation**). *Consider problem (1), where $f$ satisfies Assumption 2 (existence of a minimizer), and Assumption 5 (interpolation). Also, let each $f_i$ satisfy Assumption 1 (convexity), let Assumption 4 (contractivity) hold, and let the stepsizes $\{\gamma_k\}$ be constants conditional on $x_k$ and $w_k$. Then the iterates $\{x_k, w_k\}$ generated by Algorithm 1 (EF21-P), where $g(w_k)$ is the subgradient of $f_{i_k}$ at $w_k$, and $i_k$ is selected from $\{1, 2, \ldots, n\}$ uniformly at random, satisfy the inequality/recursion*

$$\mathrm{E}\left[V_{k+1} | x_k, w_k\right] \leq V_k - 2\mathrm{E}\left[\gamma_k[f_{i_k}(w_k) - f_{i_k}(x_\star)] | x_k, w_k\right] + B\mathrm{E}\left[\gamma_k^2 \|g(w_k)\|^2 \Big| x_k, w_k\right],$$

*where the Lyapunov function $V_k$ and constants $B$ are given by*

$$V_k := \|x_k - x_\star\|^2 + \frac{1}{\lambda(1 - (1-\alpha)(1+\theta))}\|w_k - x_k\|^2, \text{ and } B = 1 + \lambda + \frac{1}{\lambda}\frac{(1-\alpha)(1+1/\theta)}{1 - (1-\alpha)(1+\theta)}.$$

*Above, $\lambda > 0$ can be chosen arbitrarily, and $\theta > 0$ is any constant such that $(1-\alpha)(1+\theta) \in (0, 1)$.*

From this lemma, we derive the EF21-P convergence in the stochastic setting, thus leading to the same rates and discussions as in the deterministic setting in Section 5. We present the results for the constant, decreasing, and Polyak stepsizes in Theorem 4, 5 and 6 of Appendix D, respectively.

## 7    NUMERICAL EXPERIMENTS

We verify our theory by evaluating the performance of EF21-P with constant, decreasing, and Polyak stepsizes. We tested the algorithms for solving logistic regression over synthetic sparse data (see Appendix F), and support vector machines over benchmarked data from LIBSVM (Chang & Lin, 2011), as described in the next section. For all experiments, we initialized the iterates $x_0 = w_0 = 0$, and set $B = 1 + 2\lambda_\star$ with $\lambda_\star = \frac{\sqrt{1-\alpha}}{1-\sqrt{1-\alpha}}$. The results were obtained from the machine with 2.4 GHz Intel Core i5 processor, and were averaged from three Monte Carlo runs for the stochastic setting.

### 7.1    SUPPORT VECTOR MACHINE PROBLEM OVER BENCHMARKED DATA

Next, we consider the support vector machines problem, i.e. problem (1) with $f_i(x) = \max(0, 1 - b_i\langle a_i, x\rangle)$ for each dataset $\{a_i, b_i\}_{i=1}^n$. This problem is convex but nonsmooth, and the subgradient of each function is $\nabla f_i(x) = -\mathbb{1}_{b_i\langle a_i,x\rangle \leq 1} b_i a_i$, where $\mathbb{1}_{\mathcal{D}}$ is 1 if the condition $\mathcal{D}$ holds, and 0 otherwise. We ran EF21-P with the Top-$k$ sparsification, where $k$ is 5% of the problem dimension, for the problem over three benchmarked data from LIBSVM (Chang & Lin, 2011): madelon, rcv1binary, and w5a. Here, we set the constant stepsize in $\{0.2, 5.0\}$, the decreasing stepsize with $\frac{5}{\sqrt{k+1}}$, and the Polyak stepsize.

Figure 2 indicates the superior performance of using the Polyak stepsize over other stepsize schedules for EF21-P in the deterministic setting. The Polyak stepsize attains the higher solution accuracy than the constant stepsize $\gamma = 5.0$ roughly by an order of magnitude at iteration $k = 4,000$ for madelon and by four orders of magnitude at iteration $k = 2,000$ for rcv1binary.

## 8    CONCLUSION

We have presented a comprehensive analysis of EF21-P, a useful error feedback algorithm, for smooth and nonsmooth convex regimes. Our analysis allows EF21-P to employ constant, decreasing, and Polyak stepsizes for deterministic and stochastic interpolation problems. Specifically, EF21-P with the Polyak stepsizes enjoys convergence at the $\mathcal{O}(1/\sqrt{K})$ rate for nonsmooth regimes, and at the $\mathcal{O}(1/K)$ rate for smooth regimes. This marks the first successful incorporation of the Polyak stepsize into error feedback algorithms, which is useful for solving problems, where Lipschitz parameters are inaccessible. Finally, our experiments on logistic regression and support vector machines indicate significant convergence improvements achieved by EF21-P using our Polyak stepsize compared to constant and decreasing stepsizes.

**Future works.**    There are many interesting extensions of our work. By the current analysis framework in this paper, we can investigate how other adaptive stepsize strategies affect the convergence of EF21-P, e.g. Adagrad (Duchi et al., 2011), gradient diversity (Horváth et al., 2022), Armijo (Armijo, 1966), and Nonnegative Gauss-Newton stepsizes (Orvieto & Xiao, 2024). Another challenging extension of our analysis is to generalize EF21-P to minimize nonsmooth and (generalized) smooth nonconvex functions. This extension requires us to revisit the Lyapunov function or to modify the EF21-P update to prove its convergence for any stepsize strategies.

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

CONTENTS

## A    BASIC FACTS, IDENTITIES AND INEQUALITIES

We state the following useful facts from linear algebra: for any $a, b \in \mathbb{R}^d$ and any $\zeta > 0$,

$$2 \langle a, b \rangle = \|a\|^2 + \|b\|^2 - \|a - b\|^2, \tag{5}$$

$$\|a + b\|^2 \leq (1 + \zeta) \|a\|^2 + \left(1 + \zeta^{-1}\right) \|b\|^2. \tag{6}$$

Next we state the useful identity from probability. For any random variables $X, Y$, the tower property of expectation (also known as the law of iterated expectation) says that

$$\mathrm{E}\left[\mathrm{E}\left[X \mid Y\right]\right] = \mathrm{E}\left[X\right], \tag{7}$$

while the Cauchy-Schwarz inequality states that

$$\left|\mathrm{E}\left[XY\right]\right| \leq \sqrt{\mathrm{E}\left[X^2\right] \mathrm{E}\left[Y^2\right]} \tag{8}$$

## B    CONTRACTIVE COMPRESSORS AND THEIR EXAMPLES

Compressors under Assumption 4 are not assumed to be unbiased, so they can be either deterministic or stochastic. The associated contraction parameter $\alpha$ implies how close the compressed vector is to its full-precision vector (i.e., $\alpha = 1$ implies that the compressor is an identity operator). Examples of $\alpha$-contractive compressors include Top-$k$ sparsification (Alistarh et al., 2018; Stich et al., 2018), biased random sparsification (Beznosikov et al., 2023), adaptive random sparsification (Beznosikov et al., 2023), scaled-sign quantization (Karimireddy et al., 2019) and biased exponential rounding (Beznosikov et al., 2023). Table 3 summarizes examples of these $\alpha$-contractive compressors, each of which includes its type and contraction parameter.

| Compression Operator | Type | Contraction parameter $\alpha$ |
|---|---|---|
| Top-$k$ sparsification | Deterministic | $k/d$ |
| Biased random sparsification | Stochastic | $q = \min_i p_i$ |
| Adaptive random sparsification | Stochastic | $1/d$ |
| Scaled-sign quantization | Deterministic | $1/d$ |
| Biased exponential rounding | Deterministic | $4b/(1 + b)^2$ |

Table 3: Examples of $\alpha$-contractive compressors. For more examples of $\alpha$-contractive compressors, we refer to the works by Beznosikov et al. (2023), Safaryan et al. (2022b), and Albasyoni et al. (2020).

Furthermore, some compressors that may not be contractive can be made $\alpha$-contractive by scaling with a properly chosen positive scalar. In particular, $(1+\omega)^{-1}\mathcal{C}(x)$ is a $\frac{1}{1+\omega}$-contractive compressor if $\mathcal{C}(x)$ is a compressor that satisfies

$$\mathrm{E}\left[\mathcal{C}(x)\right] = x \quad \text{and} \quad \mathrm{E}\left[\|\mathcal{C}(x)\|^2\right] \leq \omega \|x\|^2 \tag{9}$$

for $\omega > 0$ and $x \in \mathbb{R}^d$. Here, we can show that $\mathrm{E}\left[\|\mathcal{C}(x) - x\|^2\right] \leq (\omega - 1)\|x\|^2$, thus implying that $\mathcal{C}(x)$ is unbiased but not $\alpha$-contractive for $\omega > 2$.

## C    PROOF OF EF21-P FOR DETERMINISTIC OPTIMIZATION

### C.1    PROOF OF LEMMA 1

We prove Lemma 1 in three steps.

**Step 1: Bound** $\mathrm{E}\left[\left.\|w_{k+1}-x_{k+1}\|^2\right|x_k,w_k\right]$**.** From the definition of the Euclidean norm,

$$
\mathrm{E}\left[\left.\|w_{k+1}-x_{k+1}\|^2\right|x_k,w_k\right] \overset{\text{Update of }w_{k+1}}{=} \mathrm{E}\left[\left.\|\mathcal{C}_k(x_{k+1}-w_k)-(x_{k+1}-w_k)\|^2\right|x_k,w_k\right]
$$

$$
\overset{(4)}{\leq} (1-\alpha)\mathrm{E}\left[\left.\|x_{k+1}-w_k\|^2\right|x_k,w_k\right]
$$

$$
\overset{\text{Update of }x_{k+1}}{=} (1-\alpha)\mathrm{E}\left[\left.\|x_k-w_k-\gamma_k g(w_k)\|^2\right|x_k,w_k\right]
$$

$$
= (1-\alpha)\|x_k-w_k-\gamma_k g(w_k)\|^2,
$$

where $g(w_k)$ is the subgradient of $f$ at $w_k$. Next, by (6), and by the fact that the stepsize $\gamma_k>0$ is conditioned on $x_k,w_k$

$$
\mathrm{E}\left[\left.\|w_{k+1}-x_{k+1}\|^2\right|x_k,w_k\right] \leq (1-\alpha)(1+\theta)\|x_k-w_k\|^2+(1-\alpha)\left(1+\frac{1}{\theta}\right)\gamma_k^2\|g(w_k)\|^2, \quad (10)
$$

where $\theta>0$ can be chosen arbitrarily.

**Step 2: Bound** $\mathrm{E}\left[\left.\|x_{k+1}-x_\star\|^2\right|x_k,w_k\right]$**.** Since $x_{k+1}=x_k-\gamma_k g(w_k)$, by expanding the square $\|x_{k+1}-x_\star\|^2$, we get

$$
\mathrm{E}\left[\left.\|x_{k+1}-x_\star\|^2\right|x_k,w_k\right] = \mathrm{E}\left[\left.\|x_k-\gamma_k g(w_k)-x_\star\|^2\right|x_k,w_k\right]
$$

$$
= \mathrm{E}\left[\left.\|x_k-x_\star\|^2-2\gamma_k\langle g(w_k),x_k-x_\star\rangle+\gamma_k^2\|g(w_k)\|^2\right|x_k,w_k\right]
$$

$$
= \|x_k-x_\star\|^2-2\gamma_k\langle g(w_k),w_k-x_\star\rangle
$$

$$
+2\gamma_k\langle g(w_k),w_k-x_k\rangle+\gamma_k^2\|g(w_k)\|^2. \quad (11)
$$

We now use the inequality $2\langle a,b\rangle=2\left\langle\sqrt{t}a,\frac{b}{\sqrt{t}}\right\rangle\leq t\|a\|^2+\frac{1}{t}\|b\|^2$, which holds for $t>0$ and all $a,b\in\mathbb{R}^d$. In particular, we use it with $a=g(w_k)$, $b=w_k-x_k$ and $t=\lambda\gamma_k$, where $\lambda>0$ is chosen arbitrarily, and obtain the bound

$$
2\gamma_k\langle g(w_k),w_k-x_k\rangle = \gamma_k\cdot 2\langle g(w_k),w_k-x_k\rangle
$$

$$
\leq \gamma_k\left(t\|g(w_k)\|^2+\frac{1}{t}\|w_k-x_k\|^2\right)
$$

$$
= \lambda\gamma_k^2\|g(w_k)\|^2+\frac{1}{\lambda}\|w_k-x_k\|^2. \quad (12)
$$

Plugging the bound (12) into (11) leads to

$$
\mathrm{E}\left[\left.\|x_{k+1}-x_\star\|^2\right|x_k,w_k\right] \leq \|x_k-x_\star\|^2-2\gamma_k\langle g(w_k),w_k-x_\star\rangle
$$

$$
+\frac{1}{\lambda}\|w_k-x_k\|^2+(1+\lambda)\gamma_k^2\|g(w_k)\|^2. \quad (13)
$$

**Step 3: Derive the descent inequality using** $V_k=\|x_k-x_\star\|^2+A\|w_k-x_k\|^2$ **with** $A>0$**.** Defining $V_k=\|x_k-x_\star\|^2+A\|w_k-x_k\|^2$ with $A>0$, we can write

$$
\mathrm{E}\left[V_{k+1}|x_k,w_k\right] \overset{(10)+\ (13)}{\leq} \|x_k-x_\star\|^2+\left(\frac{1}{\lambda}+A(1-\alpha)(1+\theta)\right)\|w_k-x_k\|^2
$$

$$
-2\gamma_k\langle g(w_k),w_k-x_\star\rangle+B\gamma_k^2\|\nabla f(w_k)\|^2,
$$

where $B=1+\lambda+A(1-\alpha)\left(1+\theta^{-1}\right)$. If we choose $\theta$ so that $(1-\alpha)(1+\theta)\in(0,1)$, and $A=\frac{1}{\lambda}\frac{1}{1-(1-\alpha)(1+\theta)}$, then $\frac{1}{\lambda}+A(1-\alpha)(1+\theta)=A$ and hence

$$
\mathrm{E}\left[V_{k+1}|x_k,w_k\right] \leq V_k-2\gamma_k\langle g(w_k),w_k-x_\star\rangle+B\gamma_k^2\|g(w_k)\|^2.
$$

Since $g(w_k)$ is the subgradient of $f$ at $w_k$, and $f$ is convex, i.e. $f(w_k)-f(x_\star)\geq\langle g(w_k),w_k-x_\star\rangle$, we get

$$
\mathrm{E}\left[V_{k+1}|x_k,w_k\right] \leq V_k-2\gamma_k\left(f(w_k)-f(x_\star)\right)+B\gamma_k^2\|g(w_k)\|^2.
$$

## C.2 PROOF OF THEOREM 1

By Theorem 1, by the tower property of expectation (7) and by choosing $\gamma_k = \gamma > 0$,

$$\mathrm{E}\left[V_{k+1}\right] \leq \mathrm{E}\left[V_k\right] - 2\gamma\mathrm{E}\left[f(w_k) - f(x_\star)\right] + B\gamma^2\mathrm{E}\left[\|g(w_k)\|^2\right]. \tag{14}$$

**Proof for the nonsmooth case.** Suppose that $f$ also satisfies Assumption 3 with $\alpha = 0$. This is equivalent to the condition that $\|g(w)\| \leq G$ with $G \geq L_0/2$ for all $w \in \mathbb{R}^d$. By (14),

$$\mathrm{E}\left[V_{k+1}\right] \leq \mathrm{E}\left[V_k\right] - 2\gamma\mathrm{E}\left[f(w_k) - f(x_\star)\right] + BG^2\gamma^2. \tag{15}$$

Next, define $\hat{w}_K = \frac{1}{K}\sum_{k=0}^{K-1} w_k$. Then, by the convexity of $f$, we get

$$
\begin{aligned}
\mathrm{E}\left[f(\hat{w}_K) - f(x_\star)\right] &\leq & \frac{1}{K}\sum_{k=0}^{K-1}\mathrm{E}\left[f(w_k) - f(x_\star)\right] \\
&\stackrel{(15)}{\leq} & \frac{\mathrm{E}\left[V_0\right] - \mathrm{E}\left[V_K\right]}{2\gamma K} + \frac{BG^2\gamma}{2} \\
&\stackrel{V_K \geq 0}{\leq} & \frac{V_0}{2\gamma K} + \frac{BG^2\gamma}{2}.
\end{aligned}
$$

**Proof for the smooth case.** Suppose that $f$ also satisfies Assumption 3 with $\alpha = 1$. By the fact that $f$ is convex and has $L_1$-Lipschitz continuous gradient, i.e

$$\|\nabla f(w_k)\|^2 \leq 2L_1\left(f(w_k) - f(x_\star)\right),$$

and by (14) we get

$$\mathrm{E}\left[V_{k+1}\right] \leq \mathrm{E}\left[V_k\right] - 2\gamma(1 - BL_1\gamma)\mathrm{E}\left[f(w_k) - f(x_\star)\right]. \tag{16}$$

If $0 < \gamma < \frac{1}{BL_1}$, then $2\gamma(1 - BL_1\gamma) > 0$. Next, by defining $\hat{w}_K = \frac{1}{K}\sum_{k=0}^{K-1} w_k$, and by the convexity of $f$,

$$
\begin{aligned}
\mathrm{E}\left[f(\hat{w}_K) - f(x_\star)\right] &\leq & \frac{1}{K}\sum_{k=0}^{K-1}\mathrm{E}\left[f(w_k) - f(x_\star)\right] \\
&\stackrel{(16)}{\leq} & \frac{\mathrm{E}\left[V_0\right] - \mathrm{E}\left[V_K\right]}{2\gamma(1 - BL_1\gamma)K} \\
&\stackrel{V_K \geq 0}{\leq} & \frac{V_0}{2\gamma(1 - BL_1\gamma)K}.
\end{aligned}
$$

## C.3 PROOF OF THEOREM 2

By Lemma 1, and by the tower property of expectation (7),

$$\mathrm{E}\left[V_{k+1}\right] \leq \mathrm{E}\left[V_k\right] - 2\gamma_k\mathrm{E}\left[f(w_k) - f(x_\star)\right] + B\gamma_k^2\mathrm{E}\left[\|\nabla f(w_k)\|^2\right]. \tag{17}$$

**Proof for the nonsmoooth case.** Suppose that $f$ also satisfies Assumption 3 with $\alpha = 0$. This is equivalent to the conditon that $\|g(w)\| \leq G$ with $G \geq L_0/2$ for all $w \in \mathbb{R}^d$. By (17),

$$\mathrm{E}\left[V_{k+1}\right] \leq \mathrm{E}\left[V_k\right] - 2\gamma_k\mathrm{E}\left[f(w_k) - f(x_\star)\right] + BG^2\gamma_k^2. \tag{18}$$

Next by defining $\bar{w}_K = \frac{1}{\sum_{k=0}^{K-1}\gamma_k}\sum_{k=0}^{K-1}\gamma_k w_k$ and by using the convexity of $f$,

$$
\begin{aligned}
f(\bar{w}_K) - f(x_\star) &\leq & \frac{1}{\sum_{k=0}^{K-1}\gamma_k}\sum_{k=0}^{K-1}\gamma_k[f(w_k) - f(x_\star)] \\
&\stackrel{(18)}{\leq} & \frac{(\mathrm{E}\left[V_0\right] - \mathrm{E}\left[V_K\right]) + (BG^2)\sum_{k=0}^{K-1}\gamma_k^2}{2\sum_{k=0}^{K-1}\gamma_k} \\
&\stackrel{V_K \geq 0}{\leq} & \frac{V_0 + (BG^2)\sum_{k=0}^{K-1}\gamma_k^2}{2\sum_{k=0}^{K-1}\gamma_k}.
\end{aligned}
$$

If $\gamma_k = \frac{\gamma_0}{\sqrt{k+1}}$ with $\gamma_0 > 0$, then

$$\sum_{k=0}^{K-1} \gamma_k \geq \frac{\gamma_0\sqrt{K}}{2}, \quad \text{and} \quad \sum_{k=0}^{K-1} \gamma_k^2 \leq 2\gamma_0^2 \log(K+1).$$

Therefore,

$$\mathrm{E}\left[f(\bar{w}_K) - f(x_\star)\right] \leq \frac{V_0 + (2BG^2)\gamma_0^2 \log(K+1)}{\gamma_0\sqrt{K}}.$$

**Proof for the smoooth case.** Suppose that $f$ also satisfies Assumption 3 with $\alpha = 1$. By the fact that $f$ is convex and has $L_1$-Lipschitz continuous gradient, i.e

$$\|\nabla f(w_k)\|^2 \leq 2L_1[f(w_k) - f(x_\star)],$$

and by (17) we have

$$\mathrm{E}\left[V_{k+1}\right] \leq \mathrm{E}\left[V_k\right] - 2\gamma_k(1 - BL_1\gamma_k)\mathrm{E}\left[f(w_k) - f(x_\star)\right].$$

If $\gamma_k = \frac{\gamma_0}{\sqrt{k+1}}$ with $\gamma_0 \in (0, 1/(2BL_1)]$, then $\gamma_k \in (0, 1/(2BL_1)]$ and

$$\mathrm{E}\left[V_{k+1}\right] \leq \mathrm{E}\left[V_k\right] - \gamma_k\mathrm{E}\left[f(w_k) - f(x_\star)\right]. \tag{19}$$

Next by defining $\hat{w}_K = \frac{1}{\sum_{k=0}^{K-1} \gamma_k} \sum_{k=0}^{K-1} \gamma_k w_k$, and by the convexity of $f$,

$$\begin{aligned}
\mathrm{E}\left[f(\bar{w}_K) - f(x_\star)\right] &\leq \frac{1}{\sum_{k=0}^{K-1} \gamma_k} \sum_{k=0}^{K-1} \gamma_k\mathrm{E}\left[f(w_k) - f(x_\star)\right] \\
&\overset{(19)}{\leq} \frac{\mathrm{E}\left[V_0\right] - \mathrm{E}\left[V_K\right]}{\sum_{k=0}^{K-1} \gamma_k} \\
&\overset{V_K \geq 0}{\leq} \frac{V_0}{\sum_{k=0}^{K-1} \gamma_k}.
\end{aligned}$$

Since $\gamma_k = \frac{\gamma_0}{\sqrt{k+1}}$ with $\gamma_0 > 0$, which yields

$$\sum_{k=0}^{K-1} \gamma_k \geq \frac{\gamma_0\sqrt{K}}{2},$$

we get

$$\mathrm{E}\left[f(\bar{w}_K) - f(x_\star)\right] \leq \frac{2V_0}{\gamma_0\sqrt{K}}.$$

## C.4 PROOF OF THEOREM 3

From Lemma 1 choose the stepsize such that

$$\gamma_k = \mathrm{argmin}_\gamma \left(V_k - 2\gamma[f(w_k) - f(x_\star)] + B\gamma^2\|g(w_k)\|^2\right) = \frac{f(w_k) - f(x_\star)}{B\|g(w_k)\|^2},$$

which hence yields

$$\mathrm{E}\left[V_{k+1}|\, x_k, w_k\right] \leq V_k - \frac{(f(w_k) - f(x_\star))^2}{B\|g(w_k)\|^2}. \tag{20}$$

**Proof for the nonsmooth case.** Suppose that $f$ also satisfies Assumption 3 with $\alpha = 0$. This is equivalent to the condition that $\|g(w)\| \leq G$ with $G \geq L_0/2$ for all $w \in \mathbb{R}^d$. Therefore, from (20)

$$\mathrm{E}\left[V_{k+1}\,|\,x_k, w_k\right] \leq V_k - \frac{1}{BG^2}\left(f(w_k) - f(x_\star)\right)^2.$$

By applying the tower property of expectation (7),

$$\mathrm{E}\left[V_{k+1}\right] \leq \mathrm{E}\left[V_k\right] - \frac{1}{BG^2}\mathrm{E}\left[(f(w_k) - f(x_\star))^2\right]. \tag{21}$$

Next define $\hat{w}_K = \frac{1}{K}\sum_{k=0}^{K-1} w_k$. Then, by the convexity of $f$,

$$\mathrm{E}\left[f(\hat{w}_K) - f(x_\star)\right] \quad \leq \quad \frac{1}{K}\sum_{k=0}^{K-1}\mathrm{E}\left[f(w_k) - f(x_\star)\right].$$

By Cauchy-Schwartz inequality (8) with $X = f(w_k) - f(x_\star)$ and $Y = 1$,

$$\mathrm{E}\left[f(\hat{w}_K) - f(x_\star)\right] \quad \leq \quad \frac{1}{\sqrt{K}}\sqrt{\sum_{k=0}^{K-1}\mathrm{E}\left[(f(w_k) - f(x_\star))^2\right]}$$

$$\overset{(21)}{\leq} \quad \frac{G\sqrt{B}}{\sqrt{K}}\sqrt{\mathrm{E}\left[V_0\right] - \mathrm{E}\left[V_K\right]}$$

$$\overset{V_K \geq 0}{\leq} \quad \frac{G\sqrt{B}}{\sqrt{K}}\sqrt{V_0}.$$

**Proof for the smooth case.** Suppose that $f$ also satisfies Assumption 3 with $\alpha = 1$. By the fact that $f(x)$ is convex and has $L_1$-Lipschitz continuous gradient, i.e

$$\|\nabla f(w_k)\|^2 \leq 2L_1[f(w_k) - f(x_\star)],$$

and next by the tower property of expectation (7) and by (20),

$$\mathrm{E}\left[V_{k+1}\right] \leq \mathrm{E}\left[V_k\right] - \frac{\mathrm{E}\left[f(w_k) - f(x_\star)\right]}{2L_1 B}. \tag{22}$$

Next define $\hat{w}_K = \frac{1}{K}\sum_{k=0}^{K-1} w_k$. Then, by the convexity of $f$,

$$\mathrm{E}\left[f(\hat{w}_K) - f(x_\star)\right] \quad \leq \quad \frac{1}{K}\sum_{k=0}^{K-1}\mathrm{E}\left[f(w_k) - f(x_\star)\right]$$

$$\overset{(22)}{\leq} \quad \frac{2L_1 B}{K}\left(\mathrm{E}\left[V_0\right] - \mathrm{E}\left[V_K\right]\right)$$

$$\overset{V_K \geq 0}{\leq} \quad \frac{2L_1 B}{K}V_0.$$

# D  THEORY OF EF21-P FOR STOCHASTIC OPTIMIZATION UNDER INTERPOLATION

In this section we provide the convergence of EF21-P for convex, stochastic optimization under interpolation regimes. The following theorems are analogous to deterministic optimization, and apply for constant, decreasing and Polyak stepsizes.

**Theorem 4** (**Fixed stepsize for stochastic problems**). *Consider the iterates $\{w_k\}$ generated by Algorithm 1 (EF21-P), where $g_k$ is the subgradient of $f_{i_k}$ at $w_k$, and $i_k$ is selected from $\{1, 2, \ldots, n\}$, for problem (1). Let the assumptions invoked in Lemma 2 hold, and choose the stepsize*

$$\gamma_k = \gamma > 0.$$

*Here, $B = 1 + \lambda + \frac{1}{\lambda}\frac{(1-\alpha)(1+1/\theta)}{1-(1-\alpha)(1+\theta)}$, $\lambda > 0$ is arbitrarily chosen, and $\theta > 0$ is any constant such that $(1 - \alpha)(1 + \theta) \in (0, 1)$.*

1. *(**Nonsmooth case**). If each $f_i$ satisfies Assumption 3 with $\alpha = 0$ (G-bounded norm of $g(w)$ with $G \geq L_0/2$), then*

$$\mathrm{E}\left[f\left(\frac{1}{K}\sum_{k=0}^{K-1}w_k\right) - f(x_\star)\right] \leq \frac{V_0}{2\gamma K} + \frac{BG^2\gamma}{2}.$$

2. *(**Smooth case**). If each $f_i$ satisfies Assumption 3 with $\alpha = 1$ ($L_1$-Lipschitzness of $\nabla f_i$), and $\gamma < \frac{1}{BL_1}$, then*

$$\mathrm{E}\left[f\left(\frac{1}{K}\sum_{k=0}^{K-1}w_k\right) - f(x_\star)\right] \leq \frac{V_0}{2\gamma(1 - BL_1\gamma)K}.$$

**Theorem 5** (**Decreasing stepsize for stochastic problems**). *Consider the iterates $\{w_k\}$ generated by Algorithm 1 (EF21-P), where $g_k$ is the subgradient of $f_{i_k}$ at $w_k$, and $i_k$ is selected from $\{1, 2, \ldots, n\}$, for solving problem (1). Let the assumptions invoked in Lemma 2 hold, and choose the stepsize*

$$\gamma_k = \frac{\gamma_0}{\sqrt{k+1}} \quad with \quad \gamma_0 > 0.$$

*Here, $B = 1 + \lambda + \frac{1}{\lambda}\frac{(1-\alpha)(1+1/\theta)}{1-(1-\alpha)(1+\theta)}$, $\lambda > 0$ is arbitrarily chosen, and $\theta > 0$ is any constant such that $(1 - \alpha)(1 + \theta) \in (0, 1)$.*

1. *(**Nonsmooth case**). If each $f_i$ satisfies Assumption 3 with $\alpha = 0$ (G-bounded norm of $g(w)$ with $G \geq L_0/2$), then*

$$\mathrm{E}\left[f\left(\frac{1}{\sum_{k=0}^{K-1}\gamma_k}\sum_{k=0}^{K-1}\gamma_k w_k\right) - f(x_\star)\right] \leq \frac{V_0 + \gamma_0^2(2BG^2)\log(K+1)}{\gamma_0\sqrt{K}}.$$

2. *(**Smooth case**). If each $f_i$ satisfies Assumption 3 with $\alpha = 1$ ($L_1$-Lipschitzness of $\nabla f_i$), and $\gamma_0 \in (0, 1/(2BL_1)]$, then*

$$\mathrm{E}\left[f\left(\frac{1}{\sum_{k=0}^{K-1}\gamma_k}\sum_{k=0}^{K-1}\gamma_k w_k\right) - f(x_\star)\right] \leq \frac{2V_0}{\gamma_0\sqrt{K}}.$$

**Theorem 6** (**Polyak stepsize for stochastic problems**). *Consider the iterates $\{w_k\}$ generated by Algorithm 1 (EF21-P), where $g_k$ is the subgradient of $f_{i_k}$ at $w_k$, and $i_k$ is selected from $\{1, 2, \ldots, n\}$ for problem (1). Let the assumptions invoked in Lemma 2 hold, and choose the stepsize*

$$\gamma_k = \frac{f_{i_k}(w_k) - f_{i_k}(x_\star)}{B\|\nabla f_{i_k}(w_k)\|^2}.$$

*Here, $B = 1 + \lambda + \frac{1}{\lambda}\frac{(1-\alpha)(1+1/\theta)}{1-(1-\alpha)(1+\theta)}$, $\lambda > 0$ is arbitrarily chosen, and $\theta > 0$ is any constant such that $(1 - \alpha)(1 + \theta) \in (0, 1)$.*

1. **(Nonsmooth case).** *If each $f_i$ satisfies Assumption 3 with $\alpha = 0$ (G-bounded norm of $g(w)$ with $G \geq L_0/2$), then*

$$\mathrm{E}\left[ f\left( \frac{1}{K} \sum_{k=0}^{K-1} w_k \right) - f(x_\star) \right] \leq \frac{G\sqrt{B}}{\sqrt{K}} \sqrt{V_0}.$$

2. **(Smooth case).** *If each $f_i$ satisfies Assumption 3 with $\alpha = 1$ ($L_1$-Lipschitzness of $\nabla f_i$), then*

$$\mathrm{E}\left[ f\left( \frac{1}{K} \sum_{k=0}^{K-1} w_k \right) - f(x_\star) \right] \leq \frac{2L_1 B}{K} V_0.$$

# E    PROOF OF EF21-P FOR STOCHASTIC OPTIMIZATION UNDER INTERPOLATION

## E.1    PROOF OF LEMMA 2

We prove Lemma 2 in three steps.

**Step 1: Bound** $\mathrm{E}\left[ \|w_{k+1} - x_{k+1}\|^2 \,\Big|\, x_k, w_k \right]$. From the definition of the Euclidean norm,

$$\mathrm{E}\left[ \|w_{k+1} - x_{k+1}\|^2 \,\Big|\, x_k, w_k \right] \overset{\text{Update for } w_{k+1}}{=} \mathrm{E}\left[ \|\mathcal{C}_k(x_{k+1} - w_k) - (x_{k+1} - w_k)\|^2 \,\Big|\, x_k, w_k \right]$$

$$\overset{(4)}{\leq} (1-\alpha)\mathrm{E}\left[ \|x_{k+1} - w_k\|^2 \,\Big|\, x_k, w_k \right]$$

$$\overset{\text{Update for } x_{k+1}}{=} (1-\alpha)\mathrm{E}\left[ \|x_k - w_k - \gamma_k g(w_k)\|^2 \,\Big|\, x_k, w_k \right],$$

where $g(w_k)$ is the subgradient of $f_{i_k}$ at $w_k$ where $i_k$ is selected from $\{1, 2, \ldots, n\}$ uniformly at random. By (6), and by the fact that the stepsize $\gamma_k > 0$ is conditioned on $x_k, w_k$

$$\mathrm{E}\left[ \|w_{k+1} - x_{k+1}\|^2 \,\Big|\, x_k, w_k \right] \leq (1-\alpha)(1+\theta)\|x_k - w_k\|^2$$

$$+ (1-\alpha)(1+1/\theta)\mathrm{E}\left[ \gamma_k^2 \|g(w_k)\|^2 \,\Big|\, x_k, w_k \right]. \quad (23)$$

**Step 2: Bound** $\mathrm{E}\left[ \|x_{k+1} - x_\star\|^2 \,\Big|\, x_k, w_k \right]$. From the definition of the Euclidean norm and by the update for $x_{k+1}$,

$$\|x_{k+1} - x_\star\|^2 = \|x_k - x_\star\|^2 - 2\gamma_k \langle g(w_k), x_k - x_\star \rangle + \gamma_k^2 \|g(w_k)\|^2$$

$$= \|x_k - x_\star\|^2 - 2\gamma_k \langle g(w_k), w_k - x_\star \rangle + 2\gamma_k \langle g(w_k), w_k - x_k \rangle + \gamma_k^2 \|g(w_k)\|^2.$$

Next, we use the inequality $2\langle a, b \rangle \leq \frac{1}{t}\|a\|^2 + t\|b\|^2$ that holds for $t > 0$ and for $a, b \in \mathbb{R}^d$. In particular, we use it with $a = w_k - x_k$, $b = g(w_k)$ and $t = \lambda\gamma_k$, and hence obtain

$$2\gamma_k \langle g(w_k), w_k - x_k \rangle = \gamma_k \cdot 2\langle g(w_k), w_k - x_k \rangle$$

$$\leq \gamma_k \left( \frac{1}{\lambda\gamma_k}\|w_k - x_k\|^2 + \lambda\gamma_k \|g(w_k)\|^2 \right)$$

$$\leq \frac{1}{\lambda}\|w_k - x_k\|^2 + \lambda\gamma_k^2 \|g(w_k)\|^2.$$

Therefore,

$$\|x_{k+1} - x_\star\|^2 \leq \|x_k - x_\star\|^2 - 2\gamma_k \langle g(w_k), w_k - x_\star \rangle$$

$$+ \frac{1}{\lambda}\|w_k - x_k\|^2 + (1+\lambda)\gamma_k^2 \|g(w_k)\|^2$$

Suppose that Assumption 5 (interpolation) and the convexity of each $f_i$ hold. By the fact that $g(w_k)$ is the subgradient of $f_{i_k}$ at $w_k$,

$$
\begin{aligned}
\langle g(w_k), w_k - x_\star \rangle &= \langle g(w_k) - g(x_\star), w_k - x_\star \rangle \\
&\geq f_{i_k}(w_k) - f_{i_k}(x_\star).
\end{aligned}
$$

Therefore,

$$
\begin{aligned}
\|x_{k+1} - x_\star\|^2 &\leq \|x_k - x_\star\|^2 - 2\gamma_k[f_{i_k}(w_k) - f_{i_k}(x_\star)] \\
&\quad + \frac{1}{\lambda}\|w_k - x_k\|^2 + (1+\lambda)\gamma_k^2 \|g(w_k)\|^2.
\end{aligned}
$$

By taking the conditional expectation on $x_k, w_k$ and by the fact that the stepsize $\gamma_k$ is conditioned on $x_k, w_k$,

$$
\begin{aligned}
\mathrm{E}\left[\|x_{k+1} - x_\star\|^2 \,\middle|\, x_k, w_k\right] &\leq \|x_k - x_\star\|^2 - 2\mathrm{E}\left[\gamma_k[f_{i_k}(w_k) - f_{i_k}(x_\star)] \,\middle|\, x_k, w_k\right] \\
&\quad + \frac{1}{\lambda}\|w_k - x_k\|^2 + (1+\lambda)\mathrm{E}\left[\gamma_k^2\|g(w_k)\|^2 \,\middle|\, x_k, w_k\right]. \quad (24)
\end{aligned}
$$

**Step 3: Derive the descent inequality using** $V_k = \|x_k - x_\star\|^2 + A\|w_k - x_k\|^2$ **with** $A > 0$.
Define $V_k = \|x_k - x_\star\|^2 + A\|w_k - x_k\|^2$ with $A > 0$. Then,

$$
\begin{aligned}
\mathrm{E}\left[V_{k+1} \,\middle|\, x_k, w_k\right] &\overset{(23)+\,(24)}{\leq} \|x_k - x_\star\|^2 + (1/\lambda + A(1-\alpha)(1+\theta))\|w_k - x_k\|^2 \\
&\quad - 2\mathrm{E}\left[\gamma_k[f_{i_k}(w_k) - f_{i_k}(x_\star)] \,\middle|\, x_k, w_k\right] + B\mathrm{E}\left[\gamma_k^2\|g(w_k)\|^2 \,\middle|\, x_k, w_k\right],
\end{aligned}
$$

where $B = 1 + \lambda + A(1-\alpha)(1+1/\theta)$.
If $(1-\alpha)(1+\theta) \in (0,1)$ and $A = \frac{1}{\lambda}\frac{1}{1-(1-\alpha)(1+\theta)}$, then $1/\lambda + A(1-\alpha)(1+\theta) = A$ and

$$
\mathrm{E}\left[V_{k+1} \,\middle|\, x_k, w_k\right] \leq V_k - 2\mathrm{E}\left[\gamma_k[f_{i_k}(w_k) - f_{i_k}(x_\star)] \,\middle|\, x_k, w_k\right] + B\mathrm{E}\left[\gamma_k^2\|g(w_k)\|^2 \,\middle|\, x_k, w_k\right].
$$

### E.2 PROOF OF THEOREM 4

We first choose $\gamma_k = \gamma > 0$. Then, from Lemma 2,

$$
\begin{aligned}
\mathrm{E}\left[V_{k+1} \,\middle|\, x_k, w_k\right] &\leq V_k - 2\gamma\mathrm{E}\left[[f_{i_k}(w_k) - f_{i_k}(x_\star)] \,\middle|\, x_k, w_k\right] + B\gamma^2\mathrm{E}\left[\|g(w_k)\|^2 \,\middle|\, x_k, w_k\right] \\
&= V_k - 2\gamma[f(w_k) - f(x_\star)] + B\gamma^2\mathrm{E}\left[\|g(w_k)\|^2 \,\middle|\, x_k, w_k\right]. \quad (25)
\end{aligned}
$$

**Proof for the nonsmoooth case.** Suppose that each $f_i$ also satisfies Assumption 3 with $\alpha = 0$. This is equivalent to the condition that $\|g(w)\| \leq G$ with $G \geq L_0/2$ for all $w \in \mathbb{R}^d$. From (25)

$$
\mathrm{E}\left[V_{k+1} \,\middle|\, x_k, w_k\right] \leq V_k - 2\gamma[f(w_k) - f(x_\star)] + BG^2\gamma^2.
$$

By the tower property of expectation (7),

$$
\mathrm{E}\left[V_{k+1}\right] \leq \mathrm{E}\left[V_k\right] - 2\gamma\mathrm{E}\left[f(w_k) - f(x_\star)\right] + BG^2\gamma^2. \quad (26)
$$

Next define $\hat{w}_K = \frac{1}{K}\sum_{k=0}^{K-1} w_k$. Then, by the convexity of $f$ (due to the convexity of each $f_i$),

$$
\begin{aligned}
\mathrm{E}\left[f(\hat{w}_K) - f(x_\star)\right] &\leq \frac{1}{K}\sum_{k=0}^{K-1}\mathrm{E}\left[f(w_k) - f(x_\star)\right] \\
&\overset{(26)}{\leq} \frac{\mathrm{E}\left[V_0\right] - \mathrm{E}\left[V_K\right]}{2\gamma K} + \frac{BG^2\gamma}{2} \\
&\overset{V_K \geq 0}{\leq} \frac{V_0}{2\gamma K} + \frac{BG^2\gamma}{2}.
\end{aligned}
$$

**Proof for the smoooth case.** Suppose that each $f_i$ also satisfies Assumption 3 with $\alpha = 1$. By Assumption 5, by the $L_1$-Lipschitz continuity of $g(w)$, and by the fact that $g(w_k)$ is the subgradient of $f_{i_k}$ at $w_k$

$$
\begin{aligned}
\|g(w_k)\|^2 &= \|g(w_k) - g(x_\star)\|^2 \\
&\leq 2L_1[f_{i_k}(w_k) - f_{i_k}(x_\star)].
\end{aligned}
$$

From (25),

$$
\begin{aligned}
\mathrm{E}\left[V_{k+1} | x_k, w_k\right] &\leq V_k - 2\gamma[f(w_k) - f(x_\star)] + 2BL_1\gamma^2 \mathrm{E}\left[f_{i_k}(w_k) - f_{i_k}(x_\star) | x_k, w_k\right] \\
&= V_k - 2\gamma[f(w_k) - f(x_\star)] + 2BL_1\gamma^2[f(w_k) - f(x_\star)].
\end{aligned}
$$

By the tower property of expectation (7),

$$
\mathrm{E}\left[V_{k+1}\right] \leq \mathrm{E}\left[V_k\right] - 2\gamma(1 - BL_1\gamma)\mathrm{E}\left[f(w_k) - f(x_\star)\right]. \tag{27}
$$

If $0 < \gamma < 1/(BL_1)$, then $2\gamma(1 - BL_1\gamma) > 0$. Next, by defining $\hat{w}_K = \frac{1}{K}\sum_{k=0}^{K-1} w_k$, and by the convexity of $f$,

$$
\begin{aligned}
\mathrm{E}\left[f(\hat{w}_K) - f(x_\star)\right] &\leq \frac{1}{K}\sum_{k=0}^{K-1}\mathrm{E}\left[f(w_k) - f(x_\star)\right] \\
&\overset{(27)}{\leq} \frac{\mathrm{E}\left[V_0\right] - \mathrm{E}\left[V_K\right]}{2\gamma(1 - BL_1\gamma)K} \\
&\overset{V_K \geq 0}{\leq} \frac{V_0}{2\gamma(1 - BL_1\gamma)K}.
\end{aligned}
$$

### E.3 Proof of Theorem 5

We prove the results for nonsmooth and smooth cases.

**Proof for the nonsmoooth case.** Suppose that $f$ also satisfies Assumption 3 with $\alpha = 0$. This is equivalent to the condition that $\|g(w)\| \leq G$ with $G \geq L_0/2$ for all $w \in \mathbb{R}^d$. If $\gamma_k = \frac{\gamma_0}{\sqrt{k+1}}$ with $\gamma_0 > 0$, then from Lemma 2,

$$
\begin{aligned}
\mathrm{E}\left[V_{k+1} | x_k, w_k\right] &\leq V_k - 2\gamma_k \mathrm{E}\left[f_{i_k}(w_k) - f_{i_k}(x_\star) | x_k, w_k\right] + BG^2\gamma_k^2 \\
&= V_k - 2\gamma_k[f(w_k) - f(x_\star)] + BG^2\gamma_k^2.
\end{aligned}
$$

By the tower property of expectation (7),

$$
\mathrm{E}\left[V_{k+1}\right] \leq \mathrm{E}\left[V_k\right] - 2\gamma_k\mathrm{E}\left[f(w_k) - f(x_\star)\right] + BG^2\gamma_k^2. \tag{28}
$$

Next by defining $\hat{w}_K = \frac{1}{\sum_{k=0}^{K-1}\gamma_k}\sum_{k=0}^{K-1}\gamma_k w_k$, and by the convexity of $f$ (due to the convexity of each $f_i$),

$$
\begin{aligned}
\mathrm{E}\left[f(\bar{w}_K) - f(x_\star)\right] &\leq \frac{1}{\sum_{k=0}^{K-1}\gamma_k}\sum_{k=0}^{K-1}\gamma_k\mathrm{E}\left[f(w_k) - f(x_\star)\right] \\
&\overset{(28)}{\leq} \frac{\mathrm{E}\left[V_0\right] - \mathrm{E}\left[V_K\right] + (BG^2)\sum_{k=0}^{K-1}\gamma_k^2}{2\sum_{k=0}^{K-1}\gamma_k} \\
&\overset{V_K \geq 0}{\leq} \frac{V_0 + (BG^2)\sum_{k=0}^{K-1}\gamma_k^2}{2\sum_{k=0}^{K-1}\gamma_k}.
\end{aligned}
$$

Since $\gamma_k = \frac{\gamma_0}{\sqrt{k+1}}$ with $\gamma_0 > 0$, which yields

$$
\sum_{k=0}^{K-1}\gamma_k \geq \frac{\gamma_0\sqrt{K}}{2}, \quad \text{and} \quad \sum_{k=0}^{K-1}\gamma_k^2 \leq 2\gamma_0^2\log(K+1),
$$

we get

$$
\mathrm{E}\left[f(\bar{w}_K) - f(x_\star)\right] \leq \frac{V_0 + \gamma_0^2(2BG^2)\log(K+1)}{\gamma_0\sqrt{K}}.
$$

**Proof for the smooth case.** Suppose each $f_i$ satisfies Assumption 3 with $\alpha = 1$. By Assumption 5, by the $L_1$-Lipschitz continuity of $g(w)$, and by the fact that $g(w_k)$ is the subgradient of $f_{i_k}$ at $w_k$

$$
\begin{aligned}
\|g(w_k)\|^2 &= \|g(w_k) - g(x_\star)\|^2 \\
&\leq 2L_1[f_{i_k}(w_k) - f_{i_k}(x_\star)].
\end{aligned}
$$

Next, if $\gamma_k = \frac{\gamma_0}{\sqrt{k+1}}$ with $\gamma_0 \in (0, 1/(2BL_1)]$, then $\gamma_k \in (0, 1/(2BL_1)]$, then from Lemma 2

$$
\begin{aligned}
\mathrm{E}\left[V_{k+1}\middle| x_k, w_k\right] &\leq V_k - 2\gamma_k(1 - BL_1\gamma_k)\mathrm{E}\left[f_{i_k}(w_k) - f_{i_k}(x_\star)\middle| x_k, w_k\right] \\
&= V_k - 2\gamma_k(1 - BL_1\gamma_k)[f(w_k) - f(x_\star)] \\
&\overset{\gamma_k \leq 1/(2BL_1)}{\leq} V_k - \gamma_k[f(w_k) - f(x_\star)].
\end{aligned}
$$

By the tower property of expectation (7),

$$
\mathrm{E}\left[V_{k+1}\right] \leq \mathrm{E}\left[V_k\right] - \gamma_k \mathrm{E}\left[f(w_k) - f(x_\star)\right]. \tag{29}
$$

Next by defining $\hat{w}_K = \frac{1}{\sum_{k=0}^{K-1} \gamma_k} \sum_{k=0}^{K-1} \gamma_k w_k$, and by the convexity of $f$ (due to the convexity of each $f_i$),

$$
\begin{aligned}
\mathrm{E}\left[f(\bar{w}_K) - f(x_\star)\right] &\leq \frac{1}{\sum_{k=0}^{K-1} \gamma_k} \sum_{k=0}^{K-1} \gamma_k \mathrm{E}\left[f(w_k) - f(x_\star)\right] \\
&\overset{(29)}{\leq} \frac{\mathrm{E}\left[V_0\right] - \mathrm{E}\left[V_K\right]}{\sum_{k=0}^{K-1} \gamma_k} \\
&\overset{V_K \geq 0}{\leq} \frac{V_0}{\sum_{k=0}^{K-1} \gamma_k}.
\end{aligned}
$$

Since $\gamma_k = \frac{\gamma_0}{\sqrt{k+1}}$ with $\gamma_0 > 0$, which yields

$$
\sum_{k=0}^{K-1} \gamma_k \geq \frac{\gamma_0 \sqrt{K}}{2},
$$

we get

$$
\mathrm{E}\left[f(\bar{w}_K) - f(x_\star)\right] \leq \frac{2V_0}{\gamma_0 \sqrt{K}}.
$$

### E.4 PROOF OF THEOREM 6

We first choose the stepsize

$$
\gamma_k = \frac{f_{i_k}(w_k) - f_{i_k}(x_\star)}{B\|g(w_k)\|^2},
$$

where $B = 1 + \lambda + \frac{1}{\lambda}\frac{(1-\alpha)(1+1/\theta)}{1-(1-\alpha)(1+\theta)}$. Then, from Lemma 2

$$
\begin{aligned}
\mathrm{E}\left[V_{k+1}\middle| x_k, w_k\right] &\leq V_k - 2\mathrm{E}\left[\gamma_k[f_{i_k}(w_k) - f_{i_k}(x_\star)]\middle| x_k, w_k\right] + B\mathrm{E}\left[\gamma_k^2\|g(w_k)\|^2\middle| x_k, w_k\right] \\
&= V_k - \mathrm{E}\left[\frac{[f_{i_k}(w_k) - f_{i_k}(x_\star)]^2}{B\|g(w_k)\|^2}\middle| x_k, w_k\right].
\end{aligned}
$$

By the tower property of expectation (7),

$$
\mathrm{E}\left[V_{k+1}\right] \leq \mathrm{E}\left[V_k\right] - \mathrm{E}\left[\frac{[f_{i_k}(w_k) - f_{i_k}(x_\star)]^2}{B\|g(w_k)\|^2}\right]. \tag{30}
$$

**Proof for the non-smooth case.** Suppose that $f$ also satisfies Assumption 3 with $\alpha = 0$. This is equivalent to the condition that $\|g(w)\| \leq G$ with $G \geq L_0/2$ for all $w \in \mathbb{R}^d$. From (30)

$$\mathrm{E}\left[V_{k+1}\right] \quad \leq \quad \mathrm{E}\left[V_k\right] - \frac{1}{BG^2}\mathrm{E}\left[\left[f_{i_k}(w_k) - f_{i_k}(x_\star)\right]^2\right]. \tag{31}$$

Next define $\hat{w}_K = \frac{1}{K}\sum_{k=0}^{K-1} w_k$. Then, by the convexity of $f$ (due to the convexity of each $f_i$),

$$\mathrm{E}\left[f(\hat{w}_K) - f(x_\star)\right] \quad \leq \quad \frac{1}{K}\sum_{k=0}^{K-1}\mathrm{E}\left[f(w_k) - f(x_\star)\right].$$

Since

$$\begin{aligned}
\mathrm{E}\left[f_{i_k}(w_k) - f_{i_k}(x_\star)\right] \quad &\overset{(7)}{=} \quad \mathrm{E}\left[\mathrm{E}\left[\left.f_{i_k}(w_k) - f_{i_k}(x_\star)\right| x_k, w_k\right]\right]\\
&= \quad \mathrm{E}\left[\frac{1}{n}\sum_{i=1}^{n} f_i(w_k) - f_i(x_\star)\right]\\
&= \quad \mathrm{E}\left[f(w_k) - f(x_\star)\right],
\end{aligned}$$

we then have

$$\mathrm{E}\left[f(\hat{w}_K) - f(x_\star)\right] \quad \leq \quad \frac{1}{K}\sum_{k=0}^{K-1}\mathrm{E}\left[f_{i_k}(w_k) - f_{i_k}(x_\star)\right].$$

By Cauchy-Schwartz inequality (8) with $X = f_{i_k}(w_k) - f_{i_k}(x_\star)$ and $Y = 1$,

$$\begin{aligned}
\mathrm{E}\left[f(\hat{w}_K) - f(x_\star)\right] \quad &\leq \quad \frac{1}{\sqrt{K}}\sqrt{\sum_{k=0}^{K-1}\mathrm{E}\left[\left(f_{i_k}(w_k) - f_{i_k}(x_\star)\right)^2\right]}\\
&\overset{(31)}{\leq} \quad \frac{G\sqrt{B}}{\sqrt{K}}\sqrt{\mathrm{E}\left[V_0\right] - \mathrm{E}\left[V_K\right]}\\
&\overset{V_K \geq 0}{\leq} \quad \frac{G\sqrt{B}}{\sqrt{K}}\sqrt{V_0}.
\end{aligned}$$

**Proof for the smooth case.** Suppose each $f_i$ satisfies Assumption 3 with $\alpha = 1$. By Assumption 5, by the $L_1$-Lipschitz continuity of $g(w)$, and by the fact that $g(w_k)$ is the subgradient of $f_{i_k}$ at $w_k$

$$\begin{aligned}
\|g(w_k)\|^2 \quad &= \quad \|g(w_k) - g(x_\star)\|^2\\
&\leq \quad 2L_1[f_{i_k}(w_k) - f_{i_k}(x_\star)].
\end{aligned}$$

Therefore, from (30)

$$\mathrm{E}\left[V_{k+1}\right] \leq \mathrm{E}\left[V_k\right] - \frac{\mathrm{E}\left[f_{i_k}(w_k) - f_{i_k}(x_\star)\right]}{2L_1 B}. \tag{32}$$

Next define $\hat{w}_K = \frac{1}{K}\sum_{k=0}^{K-1} w_k$. Then, by the convexity of $f$ (due to the convexity of each $f_i$),

$$\mathrm{E}\left[f(\hat{w}_K) - f(x_\star)\right] \quad \leq \quad \frac{1}{K}\sum_{k=0}^{K-1}\mathrm{E}\left[f(w_k) - f(x_\star)\right].$$

Since

$$\begin{aligned}
\mathrm{E}\left[f_{i_k}(w_k) - f_{i_k}(x_\star)\right] \quad &\overset{(7)}{=} \quad \mathrm{E}\left[\mathrm{E}\left[\left.f_{i_k}(w_k) - f_{i_k}(x_\star)\right| x_k, w_k\right]\right]\\
&= \quad \mathrm{E}\left[\frac{1}{n}\sum_{i=1}^{n} f_i(w_k) - f_i(x_\star)\right]\\
&= \quad \mathrm{E}\left[f(w_k) - f(x_\star)\right],
\end{aligned}$$

we then have

$$\begin{aligned}
\mathrm{E}\left[f(\hat{w}_K) - f(x_\star)\right] \quad &\leq \quad \frac{1}{K}\sum_{k=0}^{K-1}\mathrm{E}\left[f_{i_k}(w_k) - f_{i_k}(x_\star)\right]\\
&\overset{(32)}{\leq} \quad \frac{2L_1 B}{K}\left(\mathrm{E}\left[V_0\right] - \mathrm{E}\left[V_K\right]\right)\\
&\overset{V_K \geq 0}{\leq} \quad \frac{2L_1 B}{K}V_0.
\end{aligned}$$

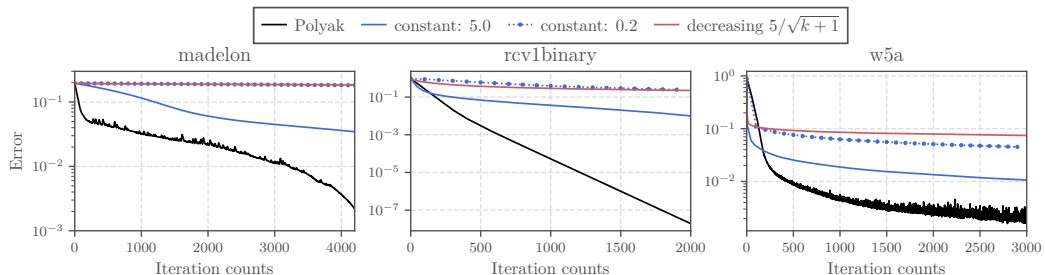

Figure 2: Convergence $f(w_k) - f(x_\star)$ w.r.t. iteration $k$ of EF21-P in the deterministic setting for the support vector machines problem.

## F  ADDITIONAL EXPERIMENTS: LOGISTIC REGRESSION PROBLEM OVER SYNTHETIC DATA

In this section, we benchmarked EF21-P for the deterministic and stochastic setting for the logistic regression, i.e. problem (1) with $f_i(x) = \log(1 + \exp(-b_i \langle a_i, x \rangle))$. This problem is convex and smooth, and the upper-bound on the Lipschitz parameter of $\nabla f_i(x)$ (and also of $\nabla f(x)$) is $\max_{i=1,2,\ldots,n} \|a_i\|^2 / 4$. We generated sparse data with the number of examples $n = 1,000$ and dimension $d = 10,000$, according to the procedure by Nutini et al. (2022). In this setup, the learning model is overparameterized (i.e. $\nabla f_i(x_\star) = 0$). In these experiments, we chose the constant stepsize $\gamma_k = \frac{1}{2BL_1}$, the decreasing stepsize $\gamma_k = \frac{1}{2BL_1\sqrt{k+1}}$, and the Polyak stepsize.

From Figure 1, the Polyak stepsize outperforms constant and decreasing stepsizes for EF21-P in the deterministic and stochastic setting. For Top-$k$ sparsification, EF21-P with the Polyak stepsize achieves higher accurate solutions than constant and decreasing stepsizes approximately by two orders of magnitude at iteration $k = 10,000$ in the deterministic setting and by an order of magnitude at $k = 25,000$ in the stochastic setting. Moreover, coarser compressors (e.g. Top-$k$ from $k = 0.1d$ to $k = 0.05d$ in Figure 1a) lead to slower convergence for EF21-P with any stepsize schedule.

