# OpenReview forum: "Error Feedback for Smooth and Nonsmooth Convex Optimization with Constant, Decreasing and Polyak Stepsizes"
_ICLR.cc/2025/Conference — Submitted to ICLR 2025_

### Official Review · Reviewer_gPkf · 2024-10-29

**Soundness:** 3
**Presentation:** 3
**Contribution:** 2
**Rating:** 3
**Confidence:** 3

**Summary:**

In this work, the authors analyzed the convergence rate of the error feedback algorithm EF21-P. The authors focused on the convex setting when the objective function can be a smooth or non-smooth convex function. Three different stepsize schema, constant, decreasing and Polyak, are considered and their convergence rates are proved. In addition, the authors provided the convergence rate when the stochastic EF21-P algorithm is applied and the objective function is a finite-sum loss function with the interpolation condition. Numerical examples are provided to support the theoretical findings.

**Strengths:**

The paper has is well-structured and easy to follow. The intuition behind the design and the theory of the new stepsize schema is clear and easy to understand. The results under the setting of distributed convex optimization should be novel (although a similar problem may have been studied under a different setting) and the proofs should be correct.

**Weaknesses:**

The major weakness of the paper is that the theory in this paper seems very preliminary. For example, in Theorem 2, the decreasing stepsize has a worse convergence rate than the constant stepsize. It would be curious if this is also the case for other schema of decreasing stepsize (e.g., $\gamma_k = O(k^{-1/3})$). Considering one specific decreasing scheme makes the theoretical contributions very limited. In addition, the authors defined Assumption 3 with a general $\eta$ but focused on two special choices, $\eta=0$ and $\eta=1$. This also seems too limited to me and may be generalized to other values of $\eta$. It would be helpful if the authors could generalize the results to other choices of decreasing stepsizes and other values of $\eta\in[0,1]$.

The authors claimed that the advantage of the Polyak stepsize is that it does not require estimating the Lipschitz constant and the optimal objective function value is easier to estimate. However, I feel that the authors did not provide convincing evidence that the optimal objective function value is easier to estimate than the Lipschitz constant, although I agree that in certain problems the optimal objective function value is exactly zero. I would suggest the authors provide more supporting evidences for this claim. For example, the authors may provide references or practical examples where the optimal value can be estimated a priori. This would help substantiate the claim about the advantages of the Polyak stepsize.

Finally, I feel that similar problem may have been studied by splitting-operator algorithms, where the compression operator $C$ can be the operator induced by the splitting operator (e.g., the Douglas–Rachford splitting or the Peaceman-Rachford splitting). It would be helpful if the authors could discuss the similarities and differences between the compression operator $C$ and a split operator. This could provide valuable context on the novelty and positioning of this work.

Overall, I feel that this paper presented some novel and interesting results, but the theory of the paper may have focused on a few very specific settings and this will limit the applicability of the results in this work.

**Questions:**

I have a few other minor comments for the authors to consider:

- The theory in Theorem 3 cannot capture the better empirical performance of Polyak stepsize than the constant stepsize. Maybe the authors could consider improving the constant before $1/\sqrt{K}$ or $1/K$.

- Lines 122 and 256: If EF21-P is equivalent to EF14, maybe the authors can choose to replace EF14 with EF21-P throughout the paper to avoid confusions.

- Line 270: $\alpha=1$ should be $\eta=1$. Also, in Theorem 1, $\alpha=0$ should be $\eta=0$.

- Assumption 3: I wonder if the assumption applies to the case when the subgradient is not unique at some points. In the case when the subgradient may not be unique, how do we choose the function $g(x)$ to guarantee the Holder continuity and the convergence? For example, in the SVM example, why did the authors choose the subgradient at line 499 when $b_{i}\langle a_{i}, x\rangle = 1$.

- Line 312: the current choice of $\lambda$ and $\theta$ only considers constant B in the inequality. This choice ignores their appearance in the Lyapunov function. I wonder if it is possible to derive the convergence rate with general $\lambda$ and $\theta$ and then choose the optimal value based on the convergence rate?

- Line 368, it should be for any initial iterates $x_0,w_0$. There are many other typos in the paper. I would suggest the authors check the paper carefully and fix the typos.

- Lines 373 and 377: $\hat{x}_{K}$ is not defined.

- Line 452: I cannot see why the interpolation condition implies that $f(x_*)=0$.

- In Figure 1, it would be better if the authors could show the performance of the constant/decreasing stepsize when they converge. It is hard to check if the current stepsize schema lead to global convergence, since it is possible that the current stepsize is too large or too small.

- In the main paper, the authors focused on the SVM example. However, the results of SVM are not presented in the main paper.

---

> ### Author Response · Authors · 2024-11-25
>
> > **The major weakness of the paper is that the theory in this paper seems very preliminary. For example, in Theorem 2, the decreasing stepsize has a worse convergence rate than the constant stepsize. It would be curious if this is also the case for other schema of decreasing stepsize (e.g., $\gamma_k = \mathcal{O}(k^{-1/3})$). Considering one specific decreasing scheme makes the theoretical contributions very limited.**
>
> We consider this comment to be minor. We agree that exploring other adaptive step size strategies, such as Adagrad, gradient diversity-based step sizes, and Nonnegative Gauss-Newton step sizes, to establish the convergence of EF21-P would further enhance our contribution.
> However, we may have misunderstood your suggestion regarding the choice of a specific decreasing step size. Using $\gamma_k = \mathcal{O}(k^{-1/3})$ ensures a convergence rate of $\mathcal{O}(K^{-2/3})$, which remains slower than the rate achieved with a properly tuned constant step size. This holds true for any stochastic gradient descent and variants. We would be happy to discuss this further.
>
>
>
>
> > **In addition, the authors defined Assumption 3 with a general $\eta$  but focused on two special choices, $\eta=0$  and $\eta=1$. This also seems too limited to me and may be generalized to other values of $\eta$. It would be helpful if the authors could generalize the results to other choices of decreasing stepsizes and other values of $\eta\in [0,1]$.**
>
> > **Assumption 3: I wonder if the assumption applies to the case when the subgradient is not unique at some points. In the case when the subgradient may not be unique, how do we choose the function $g(x)$ to guarantee the Holder continuity and the convergence? For example, in the SVM example, why did the authors choose the subgradient at line 499 when $b_i \langle a_i, x\rangle =1$. **
> **Hölder continuity:** We leverage the Hölder continuity of the subgradient, parameterized by $\eta$, to establish our convergence theorems while accommodating both nonsmooth and smooth regimes. This approach minimizes the need for restrictive assumptions about the Lipschitz continuity and smoothness of the function $f$. We concur with Reviewer 6cKe's observation that convergence theorems for gradient-based algorithms under this condition typically hold for any $\eta \in [0,1]$. This represents a compelling extension that necessitates revisiting the Lyapunov function to derive the convergence of EF21-P under general step size rules.
> **SVM:** We kindly refer you to the SVM problem discussed in Section 5 of Shamir & Zhang (2013), which provides details on computing subgradients for the regularized SVM problem. Additionally, we note a typo in the reviewer's comment: the subgradient is $-a_i b_i$ if $b_i \langle x, a_i \rangle \leq 1$, and $0$ otherwise.
>
>
> > **The authors claimed that the advantage of the Polyak stepsize is that it does not require estimating the Lipschitz constant and the optimal objective function value is easier to estimate. However, I feel that the authors did not provide convincing evidence that the optimal objective function value is easier to estimate than the Lipschitz constant, although I agree that in certain problems the optimal objective function value is exactly zero. I would suggest the authors provide more supporting evidences for this claim. For example, the authors may provide references or practical examples where the optimal value can be estimated a priori. This would help substantiate the claim about the advantages of the Polyak stepsize.**
>
> To the best of our knowledge, practical problems where the interpolation condition holds include deep neural network training problems. These problems inherently involve training over-parameterized models, as discussed in Lines 462–464.

---

> > ### Author Response · Authors · 2024-11-25
> >
> > > **Finally, I feel that similar problem may have been studied by splitting-operator algorithms, where the compression operator $C$  can be the operator induced by the splitting operator (e.g., the Douglas–Rachford splitting or the Peaceman-Rachford splitting). It would be helpful if the authors could discuss the similarities and differences between the compression operator $C$  and a split operator. This could provide valuable context on the novelty and positioning of this work.**
> >
> > The splitting operator is distinct from the contractive compressor. Specifically, the splitting operator $T$ in Douglas-Rachford splitting algorithms, e.g. [1], and other operator splitting methods assumes Lipschitz continuity. That is, there exists $L > 0$ such that for $x, y \in \mathbb{R}^d$,
> > $$\Vert T(x) - T(y) \Vert \leq L \Vert x - y \Vert.$$
> > In contrast, the contractive compressors $\mathcal{C}(v)$ we consider impose a different condition: there exists $\alpha \in (0,1)$ such that for $g \in \mathbb{R}^d$,
> > $$\Vert \mathcal{C}(g) - g \Vert^2 \leq (1-\alpha) \Vert g \Vert^2.$$
> >
> > We are happy to include this discussion in the appendix of our revised manuscript.
> > Furthermore, we believe that investigating operator splitting methods combined with error-feedback compression is another interesting future research direction.
> >
> > [1] Giselsson, Pontus. "Tight global linear convergence rate bounds for Douglas–Rachford splitting." Journal of Fixed Point Theory and Applications 19.4 (2017): 2241-2270.
> >
> > > **The theory in Theorem 3 cannot capture the better empirical performance of Polyak stepsize than the constant stepsize. Maybe the authors could consider improving the constant before $1/\sqrt{K}$ or $1/K$.**
> >
> > We believe there is a misunderstanding. Theorem 3 demonstrates that the Polyak step size achieves the same convergence performance as **carefully tuned constant step sizes**, which rely on knowledge of often inaccessible Lipschitz parameters. This result also applies to stochastic subgradient methods.
> >
> >
> > > **Line 122 and 256: If EF21-P is equivalent to EF14, maybe the authors can choose to replace EF14 with EF21-P throughout the paper to avoid confusions.**
> >
> > EF21-P, while equivalent to EF14, serves as a useful error feedback algorithm designed for server-to-client communication. Notably, EF21-P can integrate with distributed algorithms that compress client-to-server communication, such as DIANA and DCGD, enabling the creation of bi-directional compression algorithms. In contrast, EF14 focuses on saving client-to-server communication and does not support this bidirectional capability.
> >
> > > **Line 270: $\alpha=1$ should be $\eta=1$. Also, in Theorem 1, $\alpha=0$ should be $\eta=0$.**
> >
> > > **Line 368, it should be for any initial iterates $x_0,w_0$. There are many other typos in the paper. I would suggest the authors check the paper carefully and fix the typos.**
> >
> > We apologize for these typos. In the revised manuscript, we will replace $\alpha$ with $\eta$ in the convergence theorems and correct the initial iterates $x_0$ and $w_0$.

---

> > > ### Author Response · Authors · 2024-11-25
> > >
> > > > **In the main paper, the authors focused on the SVM example. However, the results of SVM are not presented in the main paper.**
> > > We consider this comment to be minor. The SVM problem is based on Shamir & Zhang (2013), where it is shown that the objective has a bounded gradient norm, which is equivalent to the Lipschitz continuity condition, and our results encompass this case.
> > >
> > > > **Line 312: the current choice of  $\lambda$ and $\theta$ only considers constant $B$ in the inequality. This choice ignores their appearance in the Lyapunov function. I wonder if it is possible to derive the convergence rate with general $\lambda$ and $\theta$, and then choose the optimal value based on the convergence rate?**
> > > We apologize for any lack of clarity in our discussion. The choice of $B_\star$ is intended to minimize $B$, as the convergence bounds in Theorem 1 with carefully tuned constant stepsizes, including Theorems 2 and 3, depend on $B$. Therefore, by choosing $B_\star$, we readily minimize $B$, which in turn minimizes the convergence bounds. We are happy to include this in the discussion.
> > >
> > >
> > > > **Lines 373 and 377: $\hat x^K$ is not defined.**
> > >
> > > $\hat w^K = \frac{1}{K}\sum_{l=0}^{K-1} w^l$. We will fix in the revised manuscript.
> > >
> > > > **Line 452: I cannot see why the interpolation condition implies that $f(x_\star)=0$.**
> > >
> > > We apologize for this typo. It must be $\nabla f(x^\star)=0$, not $f(x^\star)=0$.
> > >
> > > > **In Figure 1, it would be better if the authors could show the performance of the constant/decreasing stepsize when they converge. It is hard to check if the current stepsize schema lead to global convergence, since it is possible that the current stepsize is too large or too small.**
> > >
> > > Thank you for your comment. Due to time constraints, we are unable to provide further details. However, we expect the results to remain consistent: the Polyak step size outperforms both constant and decreasing step sizes.

---

### Official Review · Reviewer_aGTD · 2024-10-29

**Soundness:** 3
**Presentation:** 4
**Contribution:** 2
**Rating:** 8
**Confidence:** 3

**Summary:**

The paper presents new analysis for compressed (sub-) gradient descent with error feedback. The contributions include analysis for full batch GD for convex functions in the non-smooth case and analysis with decreasing and Polyak step-sizes. SGD with compression in the interpolation regime is also analyzed using a similar Lyapunov function as in the full batch case.

**Strengths:**

The paper is very well-written and easy to read. The authors invest substantial effort to explain their contribution compared to related work. This contribution, while not too original, is fair in the sense that presents analysis of a popular algorithm (error feedback) with popular choice of step-sizes apart from the constant one for which analysis already exists. In addition, there is analysis for non-smooth optimization problems (like support vector machines), which is not really surprising, but still a good to know result.

**Weaknesses:**

As discussed previously, my only issue with the paper is that it is blend of very-well known ideas, like error-feedback, adaptive step-sizes for GD, SGD in the interpolation regime etc.

**Questions:**

1. Is the Lyapunov function presented in Lemma 1 completely original, or there is some version of it in previous work?

2. It is repeatedly written over the results about Assumption 3 holding with $\alpha=0$ or $\alpha=1$, but $\alpha$ measures the quality of the compressor and Assumption 3 is about Holder continuity.

3. Why some to use diminishing step-sizes? The rate is worse and if I'm not mistaken one needs again to compute the generally unknown hyperparameters $L_0,L_1$.

4. If $f$ is not differentiable the subgradient can have many elements. What does Assumption 3 mean in this case?

---

> ### Author Response · Authors · 2024-11-25
>
> > **As discussed previously, my only issue with the paper is that it is blend of very-well known ideas, like error-feedback, adaptive step-sizes for GD, SGD in the interpolation regime etc.**
>
> We appreciate your appreciation of our work.
>
> > **Is the Lyapunov function presented in Lemma 1 completely original, or there is some version of it in previous work?**
>
> Our work builds upon related studies, including EF14 and EF21-P (Gruntkowska et al., 2023). Unlike Gruntkowska et al. (2023), which analyzed EF21-P under constant step sizes in strongly convex and smooth convex regimes, Lemmas 1 and 2 enable the analysis of EF21-P with any general step size rules, seamlessly covering both smooth and nonsmooth convex regimes.
> We also concur with Reviewer 6CKe that Lemmas 1 and 2 for EF21-P can be applied to EF14. This is because EF21-P is equivalent to EF14 in the single-node case, as demonstrated by Gruntkowska et al. (2023). However, while EF14 for nonsmooth convex regimes, as studied by Karimireddy et al. (2019), is limited to constant step sizes, Lemmas 1 and 2 extend this analysis to accommodate any general step size rules.
>
> > **It is repeatedly written over the results about Assumption 3 holding with $\alpha=0$ or $\alpha=1$, but $\alpha$ measures the quality of the compressor and Assumption 3 is about Holder continuity.**
> We apologize for these typos. In the revised manuscript, we will replace $\alpha$ with $\eta$ in the convergence theorems.
>
> > **If $f$ is not differentiable the subgradient can have many elements. What does Assumption 3 mean in this case?**
> We leverage the Hölder continuity of the subgradient, parameterized by $\eta$, to establish our convergence theorems while accommodating both nonsmooth and smooth regimes. This approach minimizes the need for restrictive assumptions about the Lipschitz continuity and smoothness of the function $f$. We concur with Reviewer 6cKe's observation that convergence theorems for gradient-based algorithms under this condition typically hold for any $\eta \in [0,1]$. This represents a compelling extension that necessitates revisiting the Lyapunov function to derive the convergence of EF21-P under general step size rules.

---

### Official Review · Reviewer_6CKe · 2024-11-01

**Soundness:** 3
**Presentation:** 2
**Contribution:** 1
**Rating:** 3
**Confidence:** 4

**Summary:**

The paper studies the error feedback mechanism under non-constant step sizes. In particular, with decreasing step sizes and Polyak step sizes. The author corroborated their theory with some numerical experiments.

**Strengths:**

The majority of the existing works on the Error Feedback mechanism focused on well-tuned constant-step sizes. This paper instead studies the case where the step sizes are changing. It also investigates the combination of EF with adaptive step sizes which seems to be the first in the field.

**Weaknesses:**

1. Seide's formulation of the Error Feedback mechanism (as is adopted in most existing works, e.g. [1]) is well-established in the literature. The authors refers to this formulation as EF14. I wonder why does the author, instead of just using this well-established formulation, use what they called "EF21-P" formulation? It seems to me that this EF21-P formulation is just writing the "virtual iteration" (used in the analysis of Seide's EF, see e.g. [1]) explicitly into the algorithm, which is a reasonable reformulation of Seide's EF but does not seem to bring any new benefit in the context of this paper. I would suggest the authors to just stick to the well-established formulation instead of rebranding it.

2. I do not understand the necessity of introducing the Hölder's smoothness condition. The paper focuses on two cases: 1) Lipschitz gradient and 2) Lipschitz continuity. Both cases are standard and extensively studied in the optimization literature. My understanding is that, when you bring up the Hölder's smoothness condition, you typically deal with "Universal methods" that automatically adapt to the best $\eta$, which is clearly not the case here. Since the theorems are stated separately for 1) and 2) anyway, I would suggest the authors to remove Assumption 3 completely to avoid confusions.

3. Regarding Lemma 1 and Lemma 2: The typical workflow of the analysis of variants of Seide's EF is the following: First you derive a descent lemma for the virtual iteration (which is the $x_t$ in this paper), and then you derive a descent lemma for the error term (which is the $w_t-x_t$ term in this paper), and then you combine the two into a Lyapunov function and proceed with the complexity analysis. Different regularity assumptions (e.g. convexity, smoothness, Lipschitzness) might affect the specific details of the derivations of these descent lemmas, but typically the overall template is not affected. The Lemma 1 and Lemma 2, though have been given strong emphasis in the main text (spanning 2 sections), seem to be a simple summary of the workflow I just described above. If these two lemmas somehow carries more insights, the authors should highlight it more clearly.

4. Regarding Section 5.1: These results seem to be well-known in the literature (see e.g. [1][2]). Is there any reason why they have to be there?

5. Regarding the changing step sizes: Typically the problem with changing stepsizes is extracted out and dealt with separately. In the case of decreasing step sizes, it seems to me that the backbone of the analysis is completely intact and one only has to do a different "summation lemma" for a complexity estimation (see, e.g. section 3.5 of [1]). Is it also the case in this paper? The case of Polyak stepsizes seems more interesting, but I would like to point out that, even though the authors claim in Table 2 that  for the Polyak step sizes the only required parameter is B, it is a bit disingenuous because you would also have to know $f^\star$.

[1] Stich, Sebastian U., and Sai Praneeth Karimireddy. "The error-feedback framework: SGD with delayed gradients." Journal of Machine Learning Research 21.237 (2020): 1-36.
[2] Sai Praneeth Karimireddy, Quentin Rebjock, Sebastian Stich, and Martin Jaggi. Error feedback fixes SGD and other gradient compression schemes. In International Conference on Machine Learning, pp. 3252–3261. PMLR, 2019.

**Questions:**

see weaknesses.

---

> ### Author Response · Authors · 2024-11-25
>
> > **Seide’s formulation of the Error Feedback mechanism (as is adopted in most existing works, e.g. [1]) is well-established in the literature. The authors refers to this formulation as EF14. I wonder why does the author, instead of just using this well-established formulation, use what they called "EF21-P" formulation? It seems to me that this EF21-P formulation is just writing the "virtual iteration" (used in the analysis of Seide's EF, see e.g. [1]) explicitly into the algorithm, which is a reasonable reformulation of Seide's EF but does not seem to bring any new benefit in the context of this paper. I would suggest the authors to just stick to the well-established formulation instead of rebranding it.**
>
> EF21-P, while equivalent to EF14, serves as a useful error feedback algorithm designed for server-to-client communication. Notably, EF21-P can integrate with distributed algorithms that compress client-to-server communication, such as DIANA and DCGD, enabling the creation of bi-directional compression algorithms. In contrast, EF14 focuses on saving client-to-server communication and does not support this bidirectional capability.
>
>
> > **Regarding Lemma 1 and Lemma 2: The typical workflow of the analysis of variants of Seide's EF is the following: First you derive a descent lemma for the virtual iteration (which is the $x_t$  in this paper), and then you derive a descent lemma for the error term (which is the $w_t-x_t$ term in this paper), and then you combine the two into a Lyapunov function and proceed with the complexity analysis. Different regularity assumptions (e.g. convexity, smoothness, Lipschitzness) might affect the specific details of the derivations of these descent lemmas, but typically the overall template is not affected. The Lemma 1 and Lemma 2, though have been given strong emphasis in the main text (spanning 2 sections), seem to be a simple summary of the workflow I just described above. If these two lemmas somehow carries more insights, the authors should highlight it more clearly.**
>
> Lemmas 1 and 2 enable us to establish convergence for error feedback algorithms using any step size strategy $\gamma_k > 0$, unlike prior works on error feedback algorithms—such as EF21-P (Gruntkowska et al., 2023) and EF14—which predominantly rely on constant step sizes to guarantee convergence. These lemmas allow us to derive the convergence of EF21-P in both smooth and nonsmooth regimes in a unified manner.
> We agree with Reviewer 6CKe that Lemmas 1 and 2 for EF21-P can be applied for EF14. This holds true because EF21-P is equivalent to EF14 for the single-node case, as shown by (Gruntkowska et al., 2023). However, EF21-P is an error feedback algorithm addressing a different issue from EF14, as explained in Concern 1.
> Moreover, one of our key contributions is presenting the first result demonstrating the successful integration of Polyak step sizes with error feedback algorithms. To the best of our knowledge, this is the first instance of adaptive step size strategies being applied to error feedback algorithms, setting it apart from prior works that only consider constant step sizes. Our framework also supports other adaptive step size methods, such as Adagrad, gradient diversity-based step sizes, and Nonnegative Gauss-Newton step sizes, for establishing convergence results.

---

> ### Author Response · Authors · 2024-11-25
>
> > **I do not understand the necessity of introducing the Hölder's smoothness condition. The paper focuses on two cases: 1) Lipschitz gradient and 2) Lipschitz continuity. Both cases are standard and extensively studied in the optimization literature. My understanding is that, when you bring up the Hölder's smoothness condition, you typically deal with "Universal methods" that automatically adapt to the best $\eta$, which is clearly not the case here. Since the theorems are stated separately for 1) and 2) anyway, I would suggest the authors to remove Assumption 3 completely to avoid confusions.**
>
>
> We leverage the Hölder continuity of the subgradient, parameterized by $\eta$, to establish our convergence theorems while accommodating both nonsmooth and smooth regimes. This approach minimizes the need for restrictive assumptions about the Lipschitz continuity and smoothness of the function $f$. We concur with Reviewer 6cKe's observation that convergence theorems for gradient-based algorithms under this condition typically hold for any $\eta \in [0,1]$. This represents a compelling extension that necessitates revisiting the Lyapunov function to derive the convergence of EF21-P under general step size rules.
>
>
>
> > **Regarding Section 5.1: These results seem to be well-known in the literature (see e.g. [1][2]). Is there any reason why they have to be there?**
>
> We show the results for Sections 5.1.-5.3 to illustrate how our descent lemmas can be used to obtain the convergence of EF21-P in both smooth and nonsmooth regimes in a unified manner. Furthermore, we derive the optimal convergence bound from Section 5.1 with constant stepsizes, which can be shown later that this bound can be achieved by the Polyak stepsize in Section 5.3.
>
>
>
>
> > **Regarding the changing step sizes: Typically the problem with changing stepsizes is extracted out and dealt with separately. In the case of decreasing step sizes, it seems to me that the backbone of the analysis is completely intact and one only has to do a different "summation lemma" for a complexity estimation (see, e.g. section 3.5 of [1]). Is it also the case in this paper? The case of Polyak stepsizes seems more interesting, but I would like to point out that, even though the authors claim in Table 2 that for the Polyak step sizes the only required parameter is $B$, it is a bit disingenuous because you would also have to know $f^{*}$.**
>
> The problem of [1] and [2] is that the analysis for EF14 is applied for constant and decreasing stepsizes. This is in contrast to our work, where the Polyak stepsize can be applied. We agree that $f^{\star}$ must be known, but it can be estimated for some classes of optimization problems, as discussed by Loizou et al. (2021).
>
> However, we agree that Lemma 14 of [1] can be utilized to derive the convergence rate from our descent lemmas in Lemmas 1 and 2.

---

> > ### Comment · Reviewer_6CKe · 2024-11-25
> >
> > I thank the authors for the response, but most of my concerns remain:
> >
> > - EF21-P vs EF14: In the context of this paper, EF21-P and EF14 seem to be exactly equivalent. The authors argued that EF21-P can be extended to accommodate bi-directional compression etc. While I don't see how EF14 cannot be extended to accommodate bi-direction compression, the important thing is that these extensions are not what's considered in this paper. Therefore, I would still suggest using EF14 instead rebranding it to EF21-P.
> >
> > - Lemma 1 and 2: In the context of constant and decreasing stepsizes, one can easily separate the analysis of 1-step descent and summation lemmas. In this way, lemma 1 and lemma 2 are essentially an incomplete 1-step descent analysis. I agree with the authors that this formulation might be beneficial if one wants to (somewhat) unify the analysis for both the decreasing stepsize and Polyak stepsize though.
> >
> > - Hölder's smoothness: does the paper contain results for $\alpha$ not equal to 1 and 0? If not, then please state them explicitly, otherwise please just stick to the 1) Lipschitz gradient and 2) Lipschitz continuity cases instead of using Hölder's smoothness which, in my opinion, only introduces unnecessary complexity.
> >
> > - Decreasing stepsizes in the literature: I agree with the authors that Lemma 14 of [1] cannot be used to derive the results for the Polyak stepsizes. However, seeing that the results of decreasing steps constitute a significant portion of the main text, it would be better if the authors can directly address and compare to Lemma 14 of [1] when discussing decreasing stepsizes.

---

### Official Review · Reviewer_Dikj · 2024-11-02

**Soundness:** 3
**Presentation:** 2
**Contribution:** 2
**Rating:** 5
**Confidence:** 2

**Summary:**

This paper studies the convergence behaviors of EF21-P when using different kind of stepsizes (fixed, decreasing, polyak) under different cases (deterministic and stochastic, convex and non-convex).

**Strengths:**

This paper presents a comprehensive study on the stepsize scheduler of EF21-P. The work contains the convergence rates of EF21-P under deterministic convex case when using constant, decreasing, and Polyak stepsize, as well as convex stochastic under an interpolation condition.

**Weaknesses:**

I find the following weaknesses in the paper and hope the author can address them:

1. The paper only focuses on the stepsize scheduler and the related convergence rates on EF-21P, however, they do not compare their convergence rates with other distributed convex optimization methods. Hence, it is quite difficult to check their theoretical contributions.

2. For the stochastic cases, the authors consider an interpolation condition which is too strong in distributed optimization, which means the optimal point of $f$ is also the optimal point of every $f_i$.


Some typos:

line 270 $\alpha=1$ should be $\eta=1$ and all left theorems use $\alpha$, which should be replaced by $\eta$.

**Questions:**

Please check the weakness parts. Also with the following questions:

1. What is challenge of dealing with the non-smoothness in analysis? It seems that doing analysis on Assumption 3, one can directly obtain a general result that cover the case of smooth $\eta=1$ and non-smooth ($\eta=0$).

I am not an expert in this field and I shall read the comments from other reviewers and the rebuttal from authors and may change my score accordingly.

---

> ### Author Response · Authors · 2024-11-25
>
> > **The paper only focuses on the stepsize scheduler and the related convergence rates on EF-21P, however, they do not compare their convergence rates with other distributed convex optimization methods. Hence, it is quite difficult to check their theoretical contributions**
>
> Unlike earlier studies on error feedback algorithms, such as EF21-P (Gruntkowska et al., 2023) and EF14, which primarily rely on constant step sizes to ensure convergence, our Lemmas 1 and 2 enable us to establish the convergence of EF21-P under any general step size rules in both smooth and nonsmooth settings in a unified framework. Additionally, we provide the first theoretical demonstration of integrating Polyak step sizes with error feedback algorithms, representing a key innovation in this paper, as highlighted by Reviewers aGTD and gPkf.
> Moreover, EF21-P can be seamlessly combined with distributed algorithms that employ compression for client-to-server communication, such as DIANA and DCGD, paving the way for bi-directional compression algorithms. While we acknowledge the suggestion to present convergence results for these bi-directional compression algorithms using EF21-P, we note that this aspect is outside the scope of the present work.
>
>
> > **For the stochastic cases, the authors consider an interpolation condition which is too strong in distributed optimization, which means the optimal point of $f$ is also the optimal point of every $f_i$.**
>
> We acknowledge that the interpolation condition can be restrictive. However, it is a widely used assumption in the analysis of gradient-based algorithms with Polyak step sizes, as demonstrated by Loizou et al. (2021) and Wang et al. (2023). This condition is crucial for proving that gradient-based algorithms with Polyak step sizes can achieve the optimal convergence bound, unlike those employing constant or decreasing step sizes.
> In contrast, for other step size strategies, such as constant or decreasing step sizes, Lemmas 1 and 2 establish the convergence of EF21-P without requiring the interpolation condition. Exploring other adaptive step size strategies for EF21-P without relying on interpolation is an intriguing direction, which we discuss in more detail in the future works section.
>
> > **line 270: $\alpha=1$ should be $\eta=1$ and all left theorems use $\alpha$, which should be replaced by $\eta$.**
>
> We apologize for these typos. In the revised manuscript, we will replace $\alpha$ with $\eta$ in Line 270, and in the convergence theorems.
>
> > **What is challenge of dealing with the non-smoothness in analysis? It seems that doing analysis on Assumption 3, one can directly obtain a general result that cover the case of smooth $\eta=1$ and nonsmooth $\eta=0$.**
>
> We leverage the Hölder continuity of the subgradient, parameterized by $\eta$, to establish our convergence theorems while accommodating both nonsmooth and smooth regimes. This approach minimizes the need for restrictive assumptions about the Lipschitz continuity and smoothness of the function $f$. We concur with Reviewer 6cKe's observation that convergence theorems for gradient-based algorithms under this condition typically hold for any $\eta \in [0,1]$. This represents a compelling extension that necessitates revisiting the Lyapunov function to derive the convergence of EF21-P under general step size rules.

---

> > ### Comment · Reviewer_Dikj · 2024-11-26
> >
> > Thanks for your response. After reading your response and the comments from other reviewers, I decide to keep my current scores. I think there are lots of places to further improve this paper:
> >
> > 1. Weaken the interpolation condition.
> >
> > 2. Give a universal analysis for H\"older continuous or just separate the condition by $\eta=0$ and $\eta=1$ as reviewer 6CKe suggests.

---

### Author Response · Authors · 2024-11-25

We appreciate all the reviewers for their appreciation of the novelty of adaptive stepsizes on error feedback algorithms under nonsmooth and smooth regimes.

We would like to start with addressing common concerns, and will address each reviewer’s comments individually.

**Concern 1: Why EF21-P, not EF14?:**   EF21-P, while equivalent to EF14, serves as a useful error feedback algorithm designed for server-to-client communication. Notably, EF21-P can integrate with distributed algorithms that compress client-to-server communication, such as DIANA and DCGD, enabling the creation of bi-directional compression algorithms. In contrast, EF14 focuses on saving client-to-server communication and does not support this bidirectional capability.

**Concern 2: Regarding Lemmas 1 and 2:**  Lemmas 1 and 2 enable us to establish convergence for error feedback algorithms using any step size strategy $\gamma_k > 0$, unlike prior works on error feedback algorithms—such as EF21-P (Gruntkowska et al., 2023) and EF14—which predominantly rely on constant step sizes to guarantee convergence. These lemmas allow us to derive the convergence of EF21-P in both smooth and nonsmooth regimes in a unified manner.

We agree with Reviewer 6CKe that Lemmas 1 and 2 for EF21-P can be applied for EF14. This holds true because EF21-P is equivalent to EF14 for the single-node case, as shown by (Gruntkowska et al., 2023). However, EF21-P is an error feedback algorithm addressing a different issue from EF14, as explained in Concern 1.

Moreover, another key contribution in our work is presenting the first result demonstrating the successful integration of Polyak step sizes with error feedback algorithms. To the best of our knowledge, this is the first instance of adaptive step size strategies being applied to error feedback algorithms, setting it apart from prior works that only consider constant step sizes. Our framework also supports other adaptive step size methods, such as Adagrad, gradient diversity-based step sizes, and Nonnegative Gauss-Newton step sizes, for establishing convergence results.

**Concern 3: Hölder's continuity of subgradient:**  We leverage the Hölder continuity of the subgradient, parameterized by $\eta$, to establish our convergence theorems while accommodating both nonsmooth and smooth regimes. This approach minimizes the need for restrictive assumptions about the Lipschitz continuity and smoothness of the function $f$. We concur with Reviewer 6cKe's observation that convergence theorems for gradient-based algorithms under this condition typically hold for any $\eta \in [0,1]$. This represents a compelling extension that necessitates revisiting the Lyapunov function to derive the convergence of EF21-P under general step size rules.

---

### Meta-Review · Area_Chair_uAsg · 2024-12-18

**Metareview:**

This work addresses limitations in the theory of error feedback, a popular method for stabilizing distributed algorithms with communication compression. It extends error feedback analysis to nonsmooth convex settings, supports constant, decreasing, and adaptive (Polyak-type) stepsizes, and introduces a combination of adaptive stepsizes with compression mechanisms. Theoretical advancements are validated with numerical experiments.

While the paper's theoretical approach is interesting, the reviewers raised several concerns regarding its novelty and assumptions. The key points are summarized below:

1. One reviewer noted the lack of comparison with other distributed convex optimization methods, making it difficult to evaluate the paper’s theoretical contributions.
2. The same reviewer criticized the use of an interpolation condition in the stochastic setting, describing it as too strong for distributed optimization.
3 Another reviewer questioned the necessity of the Hölder smoothness condition.
4. This reviewer also expressed doubts about the technical novelty of the proof techniques, pointing out that some results are already well-established in the literature.

**Additional Comments On Reviewer Discussion:**

Four reviews were collected for this paper, with three recommending rejection and one recommending acceptance. The AC agrees with the majority vote, supporting a rejection due to the unaddressed critiques raised by the reviewers. However, the AC strongly encourages resubmission, recognizing the paper's significance and potential but noting the need for substantial revisions.

---

### Decision · Program_Chairs · 2025-01-22

Reject